# Remote control of AMPK via extracellular adenosine controls tissue growth

Yao Zhang ®[1,2,3], Katrin Strassburger[1,2,3] & Aurelio A. Teleman ®[1,2,3] ✉

Adenosine monophosphate (AMP)-activated protein kinase (AMPK) is a regulator of cellular catabolism that is activated by AMP. As AMP accumulates in cells with low ATP, AMPK is considered a stress-activated kinase. While studying organ growth during *Drosophila* development, we find that AMPK can also be activated by a signalling metabolite not related to stress. Specifically, we find that two physiological inputs known to regulate organ growth rates (ecdysone (a steroid hormone) and dietary protein) modulate expression of adenosine deaminase in the intestine. This, in turn, alters circulating adenosine levels. Circulating adenosine acts as a signalling molecule by entering cells, becoming phosphorylated to AMP and activating AMPK to inhibit organ growth. Thus, AMPK activity is regulated developmentally, and AMPK activity in one tissue can be remote controlled by another tissue via circulating adenosine. Notably, this mechanism accounts for half the effect of dietary protein on tissue growth rates in *Drosophila*.

The 5′-adenosine monophosphate (AMP)-activated protein kinase (AMPK) is an important regulator of cellular homeostasis[1–7]. AMPK promotes cell catabolism and instructs cells to conserve energy by phosphorylating a large number of substrates[1–8]. To do so, AMPK inhibits mTORC1, inhibits de novo lipid biogenesis, promotes fatty acid oxidation, inhibits protein synthesis, and activates autophagy, thereby regulating almost every aspect of cell metabolism. For this reason, aberrant AMPK activity has been linked to diseases such as diabetes, cancer, and neurodegenerative disease[1,3,9].

AMPK is regulated by multiple upstream signals[4]. The most potent allosteric activator is AMP. When cells are stressed and energy is depleted, AMP levels increase, leading to AMPK activation. This yields a homeostatic loop whereby low cellular energy activates AMPK to promote catabolism and energy conservation[6]. AMPK is also activated via phosphorylation by multiple kinases including LKB1, CaMKK2 and MAP3K7, and it is inhibited by binding to glycogen[1]. Physiologically, AMPK is activated in conditions of stress, such as low glucose, calorie restriction or muscle contraction, or in conditions of inflammation such as obesity and overnutrition. Because of this, AMPK is considered a stress-activated kinase[7].

We find here that during *Drosophila* development AMPK is activated by a metabolite signalling molecule, adenosine. We find that

adenosine circulating in the blood of the animal (the haemolymph) enters cells where it is phosphorylated to adenosine monophosphate, AMP, to activate AMPK. Unlike the generation of AMP from dephosphorylation of ATP and ADP, this route of AMP synthesis is not due to stress. We find that levels of circulating adenosine are regulated both by a hormone, ecdysone, and by dietary nutrients, via regulation of expression of an enzyme in the intestine, adenosine deaminase, which degrades adenosine. This extends our understanding of AMPK beyond that of a stress induced kinase to a kinase that can also be developmentally regulated.

We discovered that adenosine can activate AMPK while studying organ growth. During development, animals undergo a dramatic increase in body mass from a single egg to the adult. Despite growing several orders of magnitude, growth terminates at a size that is prototypic for each species. The molecular mechanisms governing this robust process constitute one of the open fundamental questions in developmental biology. This question has been extensively studied using the *Drosophila* wing[10–16]. The wing is specified as a group of 30 cells in the embryo. During the three subsequent larval (instar) stages (L1 to L3) the wing grows to comprise circa 50,000 cells[14]. At this point, larval development is terminated by metamorphosis, which causes larvae to stop eating, larval tissues to histolyse and the imaginal

[1]German Cancer Research Center (DKFZ) Heidelberg, Heidelberg, Germany. [2]Faculty of Medicine, Heidelberg University, Heidelberg, Germany. [3]Faculty of Biosciences, Heidelberg University, Heidelberg, Germany. ✉e-mail: a.teleman@dkfz.de

tissues that will form adult structures, such as the wing disc, to stop proliferating[17,18]. The transitions between these developmental stages are governed by the steroid hormone ecdysone[19,20], which is functionally and structurally similar to oestrogen. Ecdysone titres increase at the end of each larval stage to induce the molt to the next stage[19–21], and they peak at the end of L3 to induce metamorphosis[17,21]. We and others previously found that ecdysone is also required for wing discs to proliferate during larval stages[18,22–29]. Therefore, in this study, we set out to better understand the requirement of ecdysone signalling for wing proliferation.

## Results

### Ecdysone autonomously promotes wing proliferation and non-autonomously represses it

To study how ecdysone influences wing proliferation, we established a system to manipulate ecdysone signalling in a spatially and temporally controlled manner. To this end, we first removed endogenous ecdysone by knocking down an enzyme required for ecdysone biosynthesis, spookier (spok). We expressed spok RNAi (spok[i])[30] using the LexA/LexAop system[31] to reserve the GAL4-UAS system for other genetic manipulations in the animal. As constitutive knockdown of ecdysone biosynthesis is early larval lethal (because ecdysone is required for every larval molt), we conditionally knocked down spok using a temperature-sensitive GAL80 (GAL80[ts]) and a temperature shift at the beginning of the last larval phase (L3). This works because the LexA driver we used has a GAL4 transactivation domain, and therefore is inhibited by GAL80 (Tub-LexA[ts]). We then assayed cell proliferation by staining wing discs with 5-ethynyl-2′-deoxyuridine (EdU), which is incorporated by cells in S-phase. This showed that wing discs no longer incorporate EdU 48 h after spok knockdown (Fig. 1a) confirming that ecdysone is required for wing proliferation, in agreement with work by us and by others both in vivo and in explant cultures[18,22–29].

We next asked where the ecdysone signalling is needed. We first activated ecdysone signalling only in wing discs. The ubiquitously expressed ecdysone receptor (EcR) is a type 2 nuclear receptor that inhibits gene expression in the absence of ligand[28,32,33]. Knockdown of EcR during larval development is sufficient to de-repress target genes, allowing wing discs to proliferate[18,27]. We therefore activated ecdysone targets by knocking down EcR only in wing discs using the C765-GAL4 driver. We did this in animals that lack endogenous ecdysone biosynthesis (Tub-LexA[ts]>spok RNAi, henceforth 'spok[i]') and therefore lack ecdysone signalling in all other tissues. As the GAL4 and LexA drivers are both sensitive to GAL80[ts], our temperature shift at early L3

simultaneously induces knockdown of spok so that the animal stops producing ecdysone, and induces knockdown of EcR in the wing so that ecdysone target genes are activated in the wing. Unlike wing discs in control spok[i] animals, which stop proliferating within 48 h after knockdown, wing discs with an EcR knockdown proliferate and remain EdU-positive indefinitely, from 48 h to the last time point we assayed, 144 h, with a visually obvious increase in size (Fig. 1b). Hence ecdysone signalling is both required and sufficient autonomously in wing discs for their proliferation.

Notably, we obtain a different outcome if we activate ecdysone target genes ubiquitously in larvae, using the Tub-GAL4 driver to knock-down EcR in all tissues. In this case, wing discs proliferate for 48 h and then terminate proliferation by 96 h (Fig. 1c). This 48-h time point (after beginning of L3) corresponds roughly to when wild-type larvae would be wandering and initiating pupation, while the 96 h time point corresponds to pupal stages in wild-type animals where wing discs no longer proliferate[17]. This transient proliferation was unexpected, given that ubiquitous EcR knockdown includes a knockdown in the wing, which should enable wing discs to proliferate indefinitely (Fig. 1b). One explanation could be that Tub-GAL4 knocks down EcR less efficiency in the wing disc compared with C765-GAL4; however, this was not the case (Extended Data Fig. 1a). To test whether a knockdown that is too strong could explain the lack of proliferation at 96 h, we combined the C765-GAL4 driver with a second wing driver, nub-GAL4, but this did not cause wing discs to stop proliferating at 96 h (Extended Data Fig. 1b). Thus, we concluded that the different outcome between Tub-GAL4 and C765-GAL4 does not result from differing knockdown efficiencies of the two drivers. Indeed, we obtained a similar result if we activated ecdysone signalling ubiquitously in a completely different manner. To do so, we fed 20-hydroxyecdysone (20E), the active form of ecdysone, to larvae lacking endogenous production (spok[i]). We previously showed that supplementing food with 25 μM 20E gives the same level of expression of ecdysone target genes as in wild-type L3 larvae[22]. Also in this case, ubiquitous activation by feeding 20E enables wing discs to proliferate at 48 h, but causes them to stop at 96 h (Fig. 1d). This occurs despite the fact that we refresh 20E in the food every 48 h to ensure that ecdysone target genes[34] remain active (Extended Data Fig. 1c).

One explanation for this discrepancy between wing disc-specific knockdown of EcR and ubiquitous knockdown of EcR is that ecdysone signalling in a tissue other than wing discs non-autonomously causes wing discs to stop proliferating at 96 h. Indeed, feeding 20E to animals with an EcR knockdown in wing discs, where wing discs cannot respond

**Fig. 1 | Ecdysone promotes wing growth autonomously and inhibits wing growth non-autonomously via enterocytes. a**, Wing discs stop proliferating within 48 h of inhibition of ecdysone synthesis, induced at the beginning of third instar via temperature shift. Wing discs, stained with EdU to assay proliferation, from control wandering L3 larvae or spookier RNAi (spok[i]) larvae. Representative images (top). Quantification of EdU (bottom). Statistical analysis by unpaired two-sided t-test. DAPI, 4,6-diamidino-2-phenylindole. **b**, Wing-specific activation of ecdysone target genes in animals otherwise lacking ecdysone signalling enables wing discs to proliferate without stopping. Lack of ecdysone is induced by knockdown of spookier (spok[i]) with a LexA driver. Wing-specific activation of ecdysone target genes is achieved with a wing-specific EcR knockdown (EcR[i]) using the C765-GAL4 driver. Discs were stained with EdU to assay proliferation at 48 h, 96 h and 144 h after shifting to 29 °C at the beginning of L3. Representative images (top). Quantification of EdU (bottom). Statistical analysis by Kruskal–Wallis test and Dunn's multiple comparisons test. **c**, Wing discs in animals lacking ecdysone (spok[i]) with ubiquitous knockdown of EcR (EcR[i]) proliferate for 48 h after RNAi induction and then terminate proliferation. Discs stained with EdU to assay proliferation at 48 h and 96 h after shifting to 29 °C. Representative images (top). Quantification of EdU (bottom). Statistical analysis by Kruskal–Wallis test and Dunn's multiple comparisons test. **d**, Wing discs from larvae lacking endogenous ecdysone biosynthesis (spok[i]) but fed with 25 μM 20E to activate ecdysone signalling in all tissues first proliferate for 48 h and then terminate

proliferation. Representative images (top). Quantification of EdU (bottom). Statistical analysis by one-way ANOVA and Dunnett's multiple comparisons test. **e**, Ecdysone reduces wing disc proliferation non-autonomously in animals with EcR knockdown in the wing. Animals lacking endogenous ecdysone synthesis (spok[i]) with C765-GAL4-driven EcR knockdown in the wing were fed with or without 25 μM 20E for 96 h. Representative images (top). Quantification of EdU (bottom). Statistical analysis by two-tailed Mann–Whitney U-test. **f**, Knockdown of EcR in enterocytes is sufficient to cause termination of wing disc proliferation at 96 h. Larvae lacking endogenous ecdysone synthesis due to spok[i], and with wing disc knockdown of EcR using C765-GAL4, which enables wing discs to proliferate indefinitely, were combined with GAL4 drivers targeting additional tissues. Representative images (top). Quantification of EdU at 96 h after RNAi induction (bottom). Statistical analysis by Kruskal–Wallis test and Dunn's multiple comparisons test. EC, enterocyte; EE, enteroendocrine cell; EB, enteroblast; ISC, intestinal stem cell. **g**, Schematic representation illustrating the ambivalent effect of ecdysone on wing disc proliferation, promoting wing disc proliferation autonomously, while inhibiting wing disc proliferation non-autonomously by acting on enterocytes in the gut. The number of biological replicates and P values are indicated in the graphs; NS, not significant. Box plots, centre line (median), box limits (first and third quartiles) and whisker (outer data points). Scale bars, 100 μm.

to ecdysone because they lack the receptor, causes the wing disc to stop proliferating at 96 h (Fig. 1e). Thus, ecdysone inhibits wing proliferation via another tissue.

## Ecdysone signalling inhibits wing proliferation via intestinal enterocytes

To identify the other tissue through which ecdysone inhibits wing proliferation, we screened a panel of drivers that express GAL4 in different

tissues such as fat body (*adh-GAL4*), muscle (*MHC-GAL4*) or the nervous system (*elav-GAL4*). We took animals that lack endogenous ecdysone synthesis (*spok*[i]) but have an *EcR* knockdown in wing discs (*C765-GAL4*), which enables wing discs to proliferate indefinitely (Fig. 1b), and then added additional GAL4 drivers to also knock down *EcR* in other tissues. This revealed that only knockdown of *EcR* with the intestinal driver *drm-GAL4* caused wing discs to stop proliferating at 96 h (Extended Data Fig. 2a,b). To identify which cell type in the intestine is responsible for

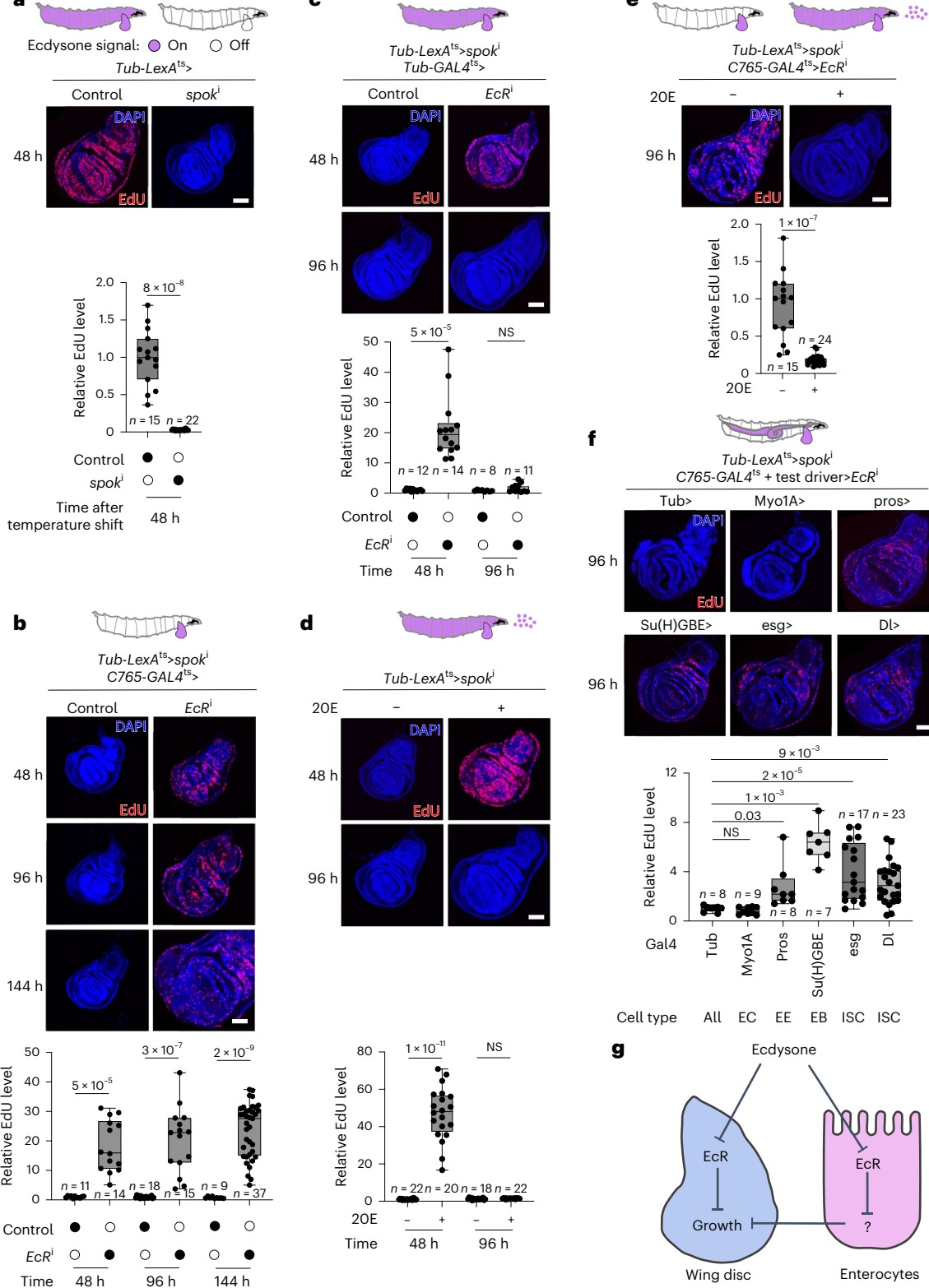

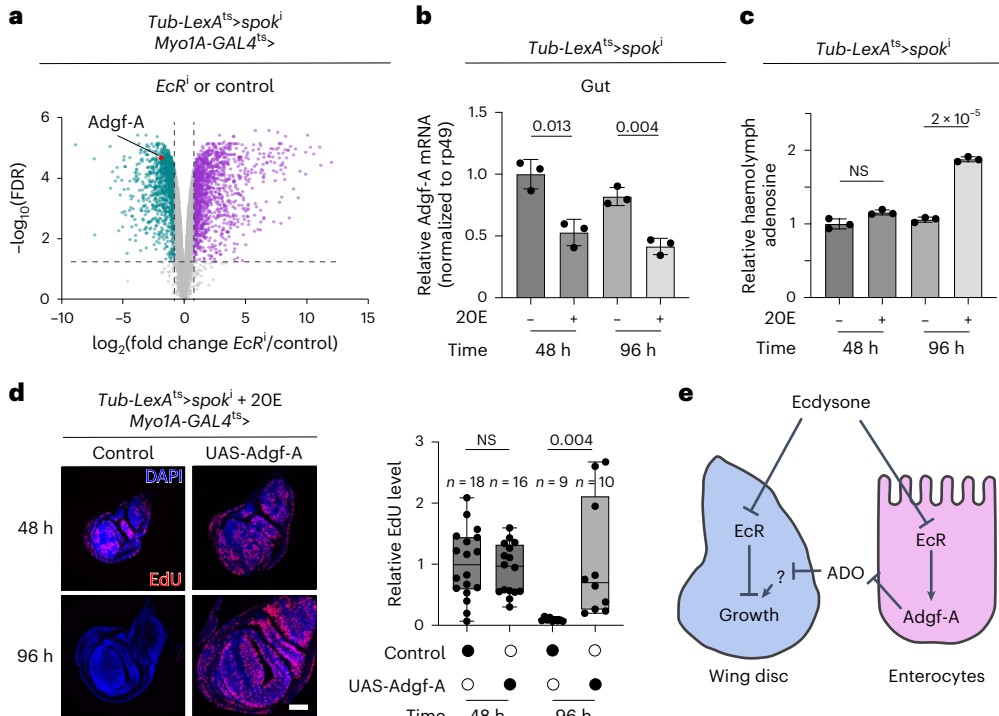

**Fig. 2 | Ecdysone inhibits wing disc proliferation by inhibiting Adgf-A expression in enterocytes, leading to accumulation of extracellular adenosine. a**, RNA-seq on intestinal RNA identifies Adgf-A as a gene downregulated upon *EcR* RNAi in enterocytes. Volcano plot of $\log_2$(fold change) versus significance. FDR was calculated with the limma-voom package on the Galaxy platform[57] with Benjamini–Hochberg adjustment. Significant candidates (FDR < 0.05) with $\log_2$(fold change) >1 are purple and <−1 are cyan. **b**, Adgf-A mRNA levels are reduced in the intestine of larvae upon 20E feeding. Adgf-A was measured by qRT–PCR on RNA from intestines from larvae lacking endogenous ecdysone synthesis (*spok$^i$*) fed with or without 25 µM 20E for 48 or 96 h. *n* = 5 larvae per sample. Representative of three biological replicates. Statistical analysis by one-way ANOVA and Dunnett's multiple comparisons test. **c**, Haemolymph adenosine levels increase 96 h after feeding larvae 20E. Larvae lack endogenous ecdysone synthesis due to *spok* RNAi. For each sample, 2 µl of

haemolymph was obtained from ten larvae at 48 h and 96 h after feeding 20E (25 µM) or vehicle. *n* = 3 biological replicates. Statistical analysis by one-way ANOVA and Dunnett's multiple comparisons test. **d**, Adgf-A expression is the limiting factor causing wing discs to stop proliferating 96 h after feeding 20E, because Adgf-A expression in the gut rescues the drop in wing proliferation at 96 h. Animals lacking endogenous ecdysone production (*spok$^i$*) and fed 25 µM 20E. Representative images (left). Quantification of EdU (right). Statistical analysis by Kruskal–Wallis test and Dunn's multiple comparisons test. The number of biological replicates is indicated. **e**, Ecdysone inhibits wing proliferation non-autonomously via enterocytes by reducing Adgf-A expression, thereby leading to elevated levels of circulating adenosine (ADO) in the haemolymph. *P* values are indicated in the graphs; NS, not significant. Bar plots, mean ± s.d. Box plot, centre line (median), box limits (first and third quartiles) and whisker (outer data points). Scale bar, 100 µm.

this effect, we tested GAL4 drivers that express specifically in intestinal enterocytes (*Myo1A-GAL4* and *Mex1-GAL4*), entero-endocrine cells (*pros-GAL4*), enteroblasts (*Su(H)GBE-GAL4*) or stem cells (*esg-GAL4* and *Dl-GAL4*) and found that *EcR* knockdown in intestinal enterocytes causes wing discs to stop proliferating at 96 h (Fig. 1f and Extended Data Fig. 2c). Both *Myo1A-GAL4* and *Mex1-GAL4* also have extremely low expression in the prothoracic gland (Extended Data Fig. 2d), but this does not explain the repression of wing disc proliferation because the prothoracic gland drivers *phm-GAL4* and *PO2O6-GAL4* do not cause this effect (Extended Data Fig. 2a,b). In sum, ecdysone has an ambivalent effect on wing disc proliferation (Fig. 1g): it autonomously promotes cell proliferation in the wing while non-autonomously inhibiting wing proliferation by acting on intestinal enterocytes. Of note, the non-autonomous inhibition occurs with a temporal delay, which we examine in more detail below.

### Ecdysone signalling inhibits wing proliferation by inhibiting Adgf-A expression in intestinal enterocytes

We next sought to understand how ecdysone signalling in intestinal enterocytes inhibits wing proliferation. As EcR is a transcription factor, we hypothesized that ecdysone regulates expression of a gene in enterocytes that affects wing proliferation. To identify this gene, we performed RNA sequencing (RNA-seq) on intestines from animals lacking endogenous ecdysone (*spok$^i$*) with or without a transient knockdown

of *EcR* to de-repress EcR targets in enterocytes using *Myo1A-GAL4$^{ts}$*. Principal-component analysis revealed that the samples clustered by genotype (Extended Data Fig. 3a), indicating consistency across biological replicates. This identified 1,346 genes that are upregulated and 1,281 genes that are downregulated upon *EcR* knockdown (false discovery rate (FDR) < 0.05; Fig. 2a and Supplementary Table 4). Gene Ontology enrichment analysis revealed that ecdysone signalling in enterocytes upregulates genes related to morphogenesis and down-regulates genes related to respiration (Extended Data Fig. 3b). Among the downregulated genes, Adgf-A caught our attention. Adgf-A is an adenosine deaminase, mainly expressed in the intestine and secreted into circulation, that converts adenosine to inosine, thereby removing adenosine from circulation (haemolymph). The Zurovec, Dolezal and Bryant laboratories have shown that Adgf-A acts as a growth factor by degrading adenosine[35–39] and we previously found that adenosine deaminase is necessary to enable wing discs to proliferate in explants[23]. This raised the hypothesis that ecdysone signalling in the intestine inhibits wing proliferation by reducing expression of Adgf-A, thereby leading to an increase in circulating adenosine. Using a green fluorescent protein insertion into the endogenous Adgf-A genomic locus[39], we confirmed that Adgf-A is expressed in all enterocytes throughout the gut (Extended Data Fig. 3c). Quantitative PCR with reverse transcription (qRT–PCR) confirmed that ecdysone signalling reduces Adgf-A expression in the gut, either in response to 20E feeding (Fig. 2b) or upon

knockdown of *EcR* in enterocytes using two independent RNAi lines (Extended Data Fig. 3d). This is followed by an increase in circulating (haemolymph) adenosine levels (Fig. 2c). Notably, there is a delay in adenosine accumulation, probably due to the fact that Adgf-A protein has a certain half-life and needs to degrade once Adgf-A mRNA levels have decreased, and then adenosine needs to accumulate once Adgf-A protein has dropped.

We performed two rescue experiments to test whether Adgf-A is the target gene responsible for blocking wing disc growth downstream of ecdysone signalling. First, we took *spok*[i] animals fed with 20E and expressed Adgf-A in intestinal enterocytes with *Myo1A-GAL4* (Fig. 2d). This revealed that elevated Adgf-A expression in intestinal enterocytes is sufficient to enable wing discs to proliferate at 96 h. Second, we took *spok*[i] animals lacking ecdysone, activated EcR signalling in wing discs with *C765-GAL4* to enable wing discs to proliferate (Extended Data Fig. 3e), and added on top the Myo1A driver to activate ecdysone signalling only in enterocytes to stop wing proliferation (Extended Data Fig. 3e). In this background, Adgf-A expression allowed wing discs to proliferate again (Extended Data Fig. 3e). Thus, ecdysone signalling in the gut inhibits wing proliferation via reduced Adgf-A expression. Together, these experiments show that the drop in intestinal Adgf-A, and the increase in adenosine, explains the non-autonomous inhibition of wing proliferation at 96 h (Fig. 2e).

### Circulating adenosine is converted to AMP to inhibit wing proliferation

We next asked how circulating adenosine inhibits wing disc proliferation. In several contexts in *Drosophila*, for instance in response to infection, adenosine acts mainly via the cell-surface adenosine receptor (AdoR)[37,40,41]. Alternatively, it has been shown in human cell culture that adenosine added exogenously to the culture medium can enter cells, become phosphorylated by adenosine kinases (Adk) to AMP and activate AMPK[42], although this mechanism has not been tested in vivo. To test these two options in vivo, we knocked down *AdoR*, *Adk2* or *Adk3* in wing discs of *spok*[i] animals fed 20E, to test which gene is required for adenosine to inhibit wing proliferation at 96 h. This showed that knockdown of *AdoR* did not enable wings to proliferate at 96 h (Fig. 3a), whereas knockdown of *Adk2* and *Adk3* did (Fig. 3b). As Adk2/3 are intracellular, this indicates that circulating adenosine needs to enter wing disc cells, and to be converted to AMP by Adk2/3 to inhibit wing proliferation (Fig. 3c). Consistent with this, AMP levels in wing discs increase at 96 h compared with 48 h (Extended Data Fig. 4a) and this was reversed upon knockdown of *Adk2/3* in the wing disc (Extended Data Fig. 4b).

### Circulating adenosine activates AMPK in the wing disc

We next asked whether circulating adenosine, which is converted to AMP, can activate AMPK in the wing disc. To test this, we manipulated the pathway shown in Fig. 3c at three different levels (EcR signalling at the top, Adgf-A expression in the middle and ADK2/3 expression at the bottom) and measured AMPK activation in wing discs, assayed via AMPK autophosphorylation. Ubiquitous activation of EcR signalling, which causes elevated adenosine levels (Fig. 2c), causes elevated AMPK activity in wing discs at 96 h post-induction compared with wing-specific knockdown of *EcR* with *C765-GAL4* (Extended Data Fig. 4c). Next, we either decreased or increased circulating adenosine by overexpressing or knocking down *Adgf-A*, respectively, in the intestine of *spok*[i] animals, and found that this leads to reduced or elevated AMPK activity in wing discs, respectively, (Fig. 4a). Finally, we tested whether the adenosine needs to be converted to AMP in wing discs to activate AMPK. Indeed, knockdown of *Adk2/3* specifically in wing discs, caused a reduction in AMPK activity (Extended Data Fig. 4d). Hence the activation of AMPK requires conversion of adenosine to AMP. This regulation of AMPK by ecdysone is analogous, but mechanistically different to what has been reported in the fat body, where ecdysone regulates transcription of AMPK subunits[43].

Of note, knockdown or overexpression of *Adgf-A* in intestinal enterocytes of animals that are otherwise wild-type also increased or decreased wing disc AMPK activity, respectively (Fig. 4b and Extended Data Fig. 4e). Thus, endogenous adenosine circulating in wild-type animals activates AMPK in imaginal discs. In sum, we conclude that circulating adenosine in larvae is imported into imaginal tissues such as the wing disc, where it is converted to AMP and activates AMPK. As a result, wing disc AMPK activity is 'remote controlled' by the intestine. Indeed, activation of ecdysone target genes in the gut using *Myo1A-GAL4* causes AMPK activation in wing discs (Extended Data Fig. 4f).

### Circulating adenosine inhibits wing proliferation via AMPK

Next we tested whether AMPK activation is the reason why wing cells stop proliferating. To address this, we first used wing disc explant cultures where we can easily use pharmacological inhibitors. We previously showed that in explants, wild-type wing discs stop proliferating if the adenosine in the medium is not removed by adding adenosine deaminase (ADA)[23] (Fig. 4c). As in larvae, adenosine in explant culture needs to enter the discs to inhibit their proliferation, as this effect is blocked by inhibiting the equilibrating nucleoside transporters Ent1, Ent2 and Ent3 with dipyridamole (Extended Data Fig. 5a) and it needs to be phosphorylated to AMP because this effect is blocked by inhibiting adenosine kinases with 5'-ITU (Fig. 4c). To test whether adenosine inhibits wing proliferation via AMPK, we explanted wing discs with an *AMPK* knockdown, and found that they can proliferate also in the absence of adenosine deaminase (Fig. 4c). From this we conclude that in explant cultures adenosine enters the disc, is converted to AMP via Adks, and activates AMPK to inhibit proliferation. To test whether this mechanism is also active in vivo, we knocked down *AMPK* in wing discs of *spok*[i] animals fed 20E, which normally stop proliferating at 96 h, and found that this enables them to continue proliferating (Fig. 4d and Extended Data Fig. 5b,c). Thus, AMPK activation is causing wing discs in these animals to stop proliferating at 96 h.

### Circulating adenosine inhibits mTORC1 to inhibit wing proliferation

As AMPK inhibits mTORC1, the data presented above raise the possibility that activation of AMPK by adenosine may lead to mTORC1 inactivation. We therefore assayed mTORC1 activity by immunoblotting wing disc lysates for RpS6 phosphorylation[44], and found that all the manipulations mentioned above, which either decrease or increase circulating adenosine, in either the *spok*[i] background or in the wild-type background, lead to an increase or a decrease in wing disc mTORC1 activity, respectively (Fig. 4a,b and Extended Data Fig. 4c–f). To test whether wing discs stop proliferating due to inactivation of mTORC1, we activated mTORC1 by knocking down the upstream inhibitor, TSC2 (ref. 45). Whereas wing discs of *spok*[i] animals fed 20E usually stop proliferating by 96 h, wing-specific activation of mTORC1 enabled the discs to continue proliferating (Fig. 4e). Conversely, inhibition of mTORC1, through knockdown of either *Rheb* or *Raptor*, reverses the proliferation enabled by *AMPK* knockdown at 96 h of 20E feeding, showing that mTORC1 acts downstream of AMPK to regulate proliferation (Extended Data Fig. 5d).

In sum, our data yield a molecular model whereby ecdysone promotes wing disc proliferation by acting directly on the wing disc, and also inhibits wing disc proliferation, but with a delay, by blocking Adgf-A expression in the gut, leading to accumulation of adenosine in circulation after some time, activation of AMPK and inhibition of mTORC1 in the wing disc (Fig. 4f).

### Adenosine levels are regulated in development

During development, ecdysone titre oscillates, peaking at every transition between larval stages and strongly increasing at the end of larval development to induce metamorphosis[21]. Hence ecdysone titre is high at the beginning of L3, decreases in the middle, and then increases again

at the end when the larvae wander. To test whether these changes affect Adgf-A expression, we quantified Adgf-A mRNA levels in intestines of early L3, mid-L3 and wandering L3 animals by qRT–PCR and found that, as expected, Adgf-A expression peaks in the middle of L3 when ecdysone titre is low, and decreases in wandering L3 when ecdysone titres increase (Fig. 5a). This is due to ecdysone signalling, because knockdown of ecdysone synthesis causes Adgf-A levels to stay high (Fig. 2b). Quantification of adenosine levels revealed that they increase at 0 and 24 h after pupariation, that is with a short delay compared with when Adgf-A levels drop in wandering animals (Fig. 5b).

### Adenosine regulates growth rate and mildly affects final organism size

We next asked whether this mechanism affects organismal growth and development. Reduction of circulating adenosine levels by Adgf-A overexpression in intestinal enterocytes causes larvae to grow faster and larger (Extended Data Fig. 6a) and to pupate faster (Fig. 5c). Hence, the endogenous levels of adenosine circulating in wild-type animals slows down their growth and development. Conversely, knockdown of Adgf-A causes a pupation delay (Fig. 5c). We then assayed adult wing size as a proxy for total body size. Animals with reduced or elevated adenosine levels have mildly increased or decreased wing size, respectively (Fig. 5d). As Adgf-A overexpression yields larger animals (Fig. 5d) that developed faster (Fig. 5c), it must cause the tissues to grow substantially faster. Consistent with this, knockdown of EcR in intestinal enterocytes, which causes reduced Adgf-A expression, causes a pupation delay that is rescued by Adgf-A overexpression (Extended Data Fig. 6b). In sum, endogenous adenosine that is in circulation acts as a brake to reduce the rate of tissue growth. We did not observe changes in the proportions of adult organs, indicating that adenosine equally represses growth of all imaginal discs. We also tested knockdown of AMPK or Adk2 + Adk3 in the wing during L3, to prevent adenosine from acting on the wing, and this led to adult wings that were larger than controls (Fig. 5e), consistent with adenosine affecting wing size via AMP and AMPK.

### Dietary yeast regulates developmental rate in part via adenosine

We next asked whether Adgf-A expression in the gut is regulated by other physiological inputs besides ecdysone. As the gut is the primary site of nutrient sensing, we tested if Adgf-A expression is regulated by dietary nutrients. We placed larvae on basal food made of PBS + non-digestible agar and supplemented it with a source of amino acids (yeast), glucose or lipid (coconut oil). qRT–PCR revealed a significant increase in Adgf-A expression in response to yeast (Fig. 5f) or amino acids (Extended Data Fig. 6c). The sensing mechanism is independent of ecdysone signalling because yeast does not reduce expression of ecdysone target genes in the gut (Extended Data Fig. 6d), and dietary yeast can increase Adgf-A expression also in spok^i animals lacking

ecdysone ('-20E'; Extended Data Fig. 6c,e). This sensing is also independent of mTORC1 because constitutive activation of mTORC1 by knocking down TSC2 or inhibition of mTORC1 by knocking down Raptor or Rheb did not blunted the induction of Adgf-A expression in response to dietary amino acids (Extended Data Fig. 6f). Instead, knockdown of either GCN2, which senses amino acid deprivation, or the downstream transcription factor ATF4 abolished the response (Extended Data Fig. 6g). Thus, dietary amino acids regulate Adgf-A expression in enterocytes via the GCN2-ATF4 pathway. In sum, Adgf-A expression

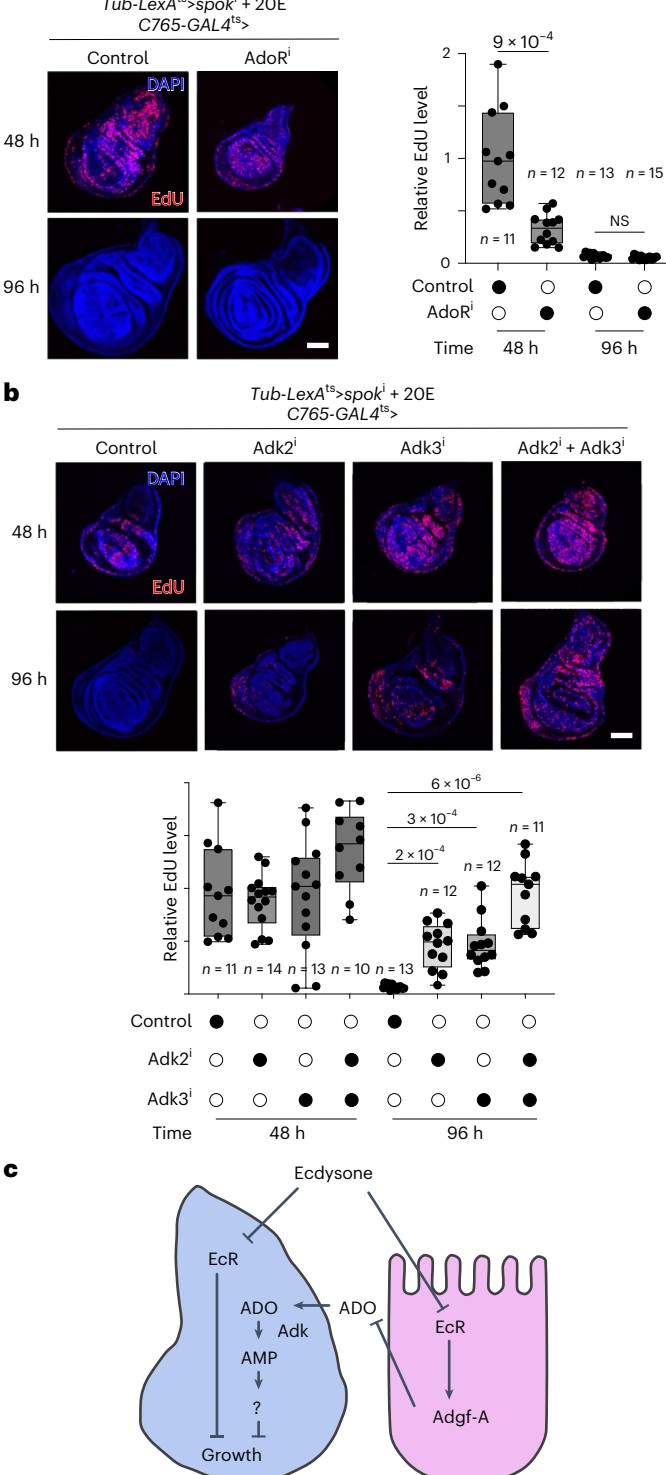

**Fig. 3 | Extracellular adenosine terminates wing disc proliferation via Adk in wing discs. a**, Knockdown of adenosine receptor (AdoR) in wing discs does not rescue the termination of wing proliferation that occurs at 96 h after feeding 20E (25 μM) to animals lacking endogenous ecdysone biosynthesis (spok^i). Representative images (left). Quantification of EdU (right). Statistical analysis by one-way ANOVA and Dunnett's multiple comparisons test. **b**, Knockdown of adenosine kinases Adk2, Adk3 or both combined in wing discs rescues the termination of wing proliferation that occurs at 96 h after feeding 20E (25 μM) to animals lacking endogenous ecdysone biosynthesis (spok^i). Representative images (top). Quantification of EdU (bottom). Statistical analysis by one-way ANOVA and Dunnett's multiple comparisons test. **c**, Schematic diagram illustrating that extracellular adenosine inhibits wing proliferation by being converted to AMP via Adk. The number of biological replicates and P values are indicated in the graphs; NS, not significant. Box plots, centre line (median), box limits (first and third quartiles) and whiskers (outer data points). Scale bars, 100 μm.

in the gut is regulated by an endogenous signal (ecdysone) and by an exogenous signal (dietary nutrients).

Dietary yeast enables *Drosophila* to develop faster[46]. As dietary yeast regulates Adgf-A expression (Fig. 5f), and Adgf-A regulates growth rate (Fig. 5c,d), this raises the possibility that dietary yeast regulates organismal growth in part via Adgf-A. To test this, we measured pupation rates of larvae placed on food with either low or high yeast. As expected, control animals on a low-yeast diet pupate slower than control animals on high yeast (Fig. 5g). Note that the high yeast food used in this experiment is richer than the standard food we used in all other experiments, including the one in Fig. 5c (Extended Data Fig. 6h). On high yeast, where Adgf-A levels are elevated (Fig. 5f), overexpression of Adgf-A had no effect on pupation timing (Fig. 5g). Overexpression of Adgf-A, however, rescued roughly half of the developmental delay caused by the low-yeast diet (Fig. 5g) indicating that the delay caused by low yeast is due in part to elevated adenosine levels. A two-way analysis of variance (ANOVA) on pupation curves from these two genotypes on three different yeast concentrations (Extended Data Fig. 6i) revealed a significant interaction between genotype and diet ($P < 0.0001$). Consistent with this effect working via AMPK and mTORC1, a low-yeast diet causes increased AMPK activity and reduced mTORC1 activity in wing discs, and this effect is blunted by Adgf-A expression (Extended Data Fig. 7a). Thus, the effect of dietary yeast on developmental rate is not only due to the nutrients contained in the yeast, but also partly due to a sensing mechanism whereby the gut senses the yeast, activates Adgf-A expression, reduces circulating adenosine and thereby accelerates the growth rate of all imaginal tissues.

### Dietary adenosine regulates growth

We next investigated whether dietary adenosine can affect organismal growth rates by supplementing the food with 1 mM adenosine. Immunoblotting of wing disc lysates revealed that dietary adenosine activates AMPK and inhibits mTORC1 in the wing (Extended Data Fig. 7b). Dietary adenosine inhibits mTORC1 via AMPK because *AMPK* knockdown rescues mTORC1 activity (Extended Data Fig. 7c). As expected, dietary adenosine causes a significant delay in the pupation rate of the animals (Extended Data Fig. 7d). This is not due to a direct effect of adenosine on ecdysone signalling, assayed by qRT–PCR on ecdysone target genes upon 24 h of adenosine feeding (Extended Data Fig. 7e). In sum, we find that circulating adenosine integrates multiple inputs to regulate developmental growth rates, including ecdysone levels, dietary yeast levels and dietary adenosine levels (Extended Data Fig. 7f).

### Circuitry logic of the ambivalent effect of ecdysone on growth

Notably ecdysone has an ambivalent effect on wing growth, both activating it autonomously, and inactivating it non-autonomously with

a delay. This circuitry could enable a pulse of ecdysone to induce a discrete amount of wing growth by inducing wing proliferation on the one hand, while on the other hand limiting the amount of time wing cells have to proliferate. This might be relevant for normal development where ecdysone levels oscillate, peaking at every larval transition, such that each peak enables organs to grow a certain amount. Although this concept will require further study, we did a proof-of-principle experiment with the *spok*[i] system to simulate two subsequent pulses of ecdysone signalling. As shown in Fig. 1d, when *spok*[i] animals lacking endogenous ecdysone are fed 20E, this causes wing discs to proliferate for 48 h and then to stop by 96 h. If we continue feeding these larvae 20E beyond 96 h for multiple days, the wing discs remain continuously EdU negative ('20E continuous'; Extended Data Fig. 8a). If instead, after 96 h of 20E feeding, we remove ecdysone from the food for 48 h and then give them a second pulse of ecdysone, the wing discs again proliferate transiently, and then stop again ('re-add'; Extended Data Fig. 8a). Thus, two subsequent pulses of ecdysone cause two discrete periods of organ growth. In wild-type larvae, the ecdysone pulses happen frequently (every 24 h) so that a complete stop in wing proliferation between pulses is not seen, nonetheless the accumulation of adenosine could slow down proliferation (Fig. 5c). In sum, this incoherent feedforward loop[47] downstream of ecdysone seems to enable ecdysone signalling to induce a discrete amount of tissue growth.

## Discussion

The data presented here support a model whereby ecdysone signalling and dietary protein regulate wing growth by controlling expression of Adgf-A in intestinal enterocytes. This affects the levels of circulating adenosine, which enters into tissues such as the wing, is converted to AMP, activates AMPK and thereby inhibits wing proliferation.

These results expand our understanding of AMPK in three ways. First, we find that AMPK can be activated by a signalling metabolite, adenosine, which is not due to stress. Instead, adenosine levels are developmentally regulated via controlled deamination downstream of ecdysone signalling. Furthermore, both flies and mammals express extracellular phosphatases that can produce adenosine via dephosphorylation of adenosine nucleotides[48,49]. Regulation of AMPK via adenosine may be relevant in multiple different aspects of physiology and disease, as AMPK regulates cellular and organismal energy balance, and AMPK is dysregulated in diabetes and cancer[3]. Second, in this context, AMPK is not sensing an intracellular signal, but a non-autonomous, extracellular signal (adenosine) that is being produced by other tissues in the body. Thus, AMPK in one tissue is being 'remote controlled' by another tissue. Finally, we find an additional physiological function of AMPK, which is to regulate the rate of organ growth. It will be interesting

**Fig. 4 | Extracellular adenosine terminates wing disc proliferation by activating AMPK. a**, Circulating adenosine regulates AMPK activation in wing discs in animals lacking endogenous ecdysone production (*spok*[i]) and fed 25 µM 20E for 96 h. Circulating adenosine levels were increased or reduced by knocking down or overexpressing *Adgf-A* in intestinal enterocytes, respectively. Representative immunoblots (top). Quantification of 4 replicates for AMPK activity (pAMPK/AMPK) and 3 replicates for mTOR activity (pS6/tubulin), 20 wing discs per condition (bottom). Statistical analysis by paired two-sided *t*-test. **b**, Circulating adenosine levels regulate wing AMPK and mTORC1 activity, assayed by either overexpressing or knocking down *Adgf-A* in enterocytes (with *Myo1A-GAL4*) in animals that are otherwise wild type. Activities are assayed by immunoblotting wing disc lysates. Representative immunoblot (top). Quantification of 4 replicates, 20 wing discs per sample (bottom). Statistical analysis by paired two-sided *t*-test. **c**, Adenosine inhibits wing disc proliferation via Adk and AMPK in explants. Control and *AMPK* knockdown wing discs cultured for 3 h ex vivo in medium containing calf intestinal ADA, Adk inhibitor 5′-ITU or neither for 3 h and then stained with EdU. Representative images (top). Quantification of EdU (bottom). Statistical analysis by Kruskal–Wallis test and Dunn's multiple comparisons test. **d**, Knockdown of *AMPK* in wing

discs of animals lacking endogenous ecdysone biosynthesis (*spok*[i]) but fed 25 µM 20E enables wing discs to continue proliferating at 96 h, showing that AMPK activation causes termination of wing proliferation. Representative images (top). Quantification of EdU (bottom). Statistical analysis by one-way ANOVA and Dunnett's multiple comparisons test. **e**, Knockdown of *TSC2* in wing discs to activate mTORC1 enables wing discs in animals lacking endogenous ecdysone biosynthesis (*spok*[i]) but fed 25 µM 20E to continue proliferating at 96 h. Representative images (top). Quantification EdU (bottom). Statistical analysis by unpaired two-tailed *t*-test. **f**, Schematic diagram summarizing the molecular mechanism by which ecdysone ambivalently regulates wing proliferation. On the one hand, ecdysone autonomously promotes proliferation in the wing disc. On the other hand, it non-autonomously inhibits wing growth with a delay by acting on intestinal enterocytes to inhibit Adgf-A expression, causing levels of circulating adenosine to increase with a delay, which then gets imported into the wing disc and phosphorylated by Adk to AMP to activate AMPK and inhibit mTORC1, thereby inhibiting proliferation. The number of biological replicates for EdU and *P* values for all are indicated in the graphs; NS, not significant. Bar plots, mean ± s.d. Box plots: centre line (median), box limits (first and third quartiles) and whisker (outer data points). Scale bars, 100 µm.

to study whether this function is evolutionarily conserved. Indeed, single-nucleotide polymorphisms in the genes encoding for ADK and the human orthologue of Adgf-A, adenosine deaminase (ADA2), are significantly associated with body height in humans[50].

Nutrients are known to regulate the growth rate of animals[46]. One nutrient frequently given to *Drosophila* to boost it growth is yeast. We find that roughly half of the effect of yeast on organismal growth rate is not directly due to its metabolic contribution to cell anabolism, but rather to its ability to activate a signalling pathway involving Adgf-A, adenosine and AMPK. This therefore uncovers a mechanism how dietary yeast controls *Drosophila* growth. Of note, it was previously shown that dietary amino acids also stimulate mTORC1 signalling in the fat body, which stimulates insulin/dILP secretion, which in turn stimulates growth of all tissues in the body[51–54]. Thus, it seems that nutrients mainly boost organ growth and developmental rate by activating signalling pathways that sense the nutrients, rather than by directly impinging upon metabolic flux.

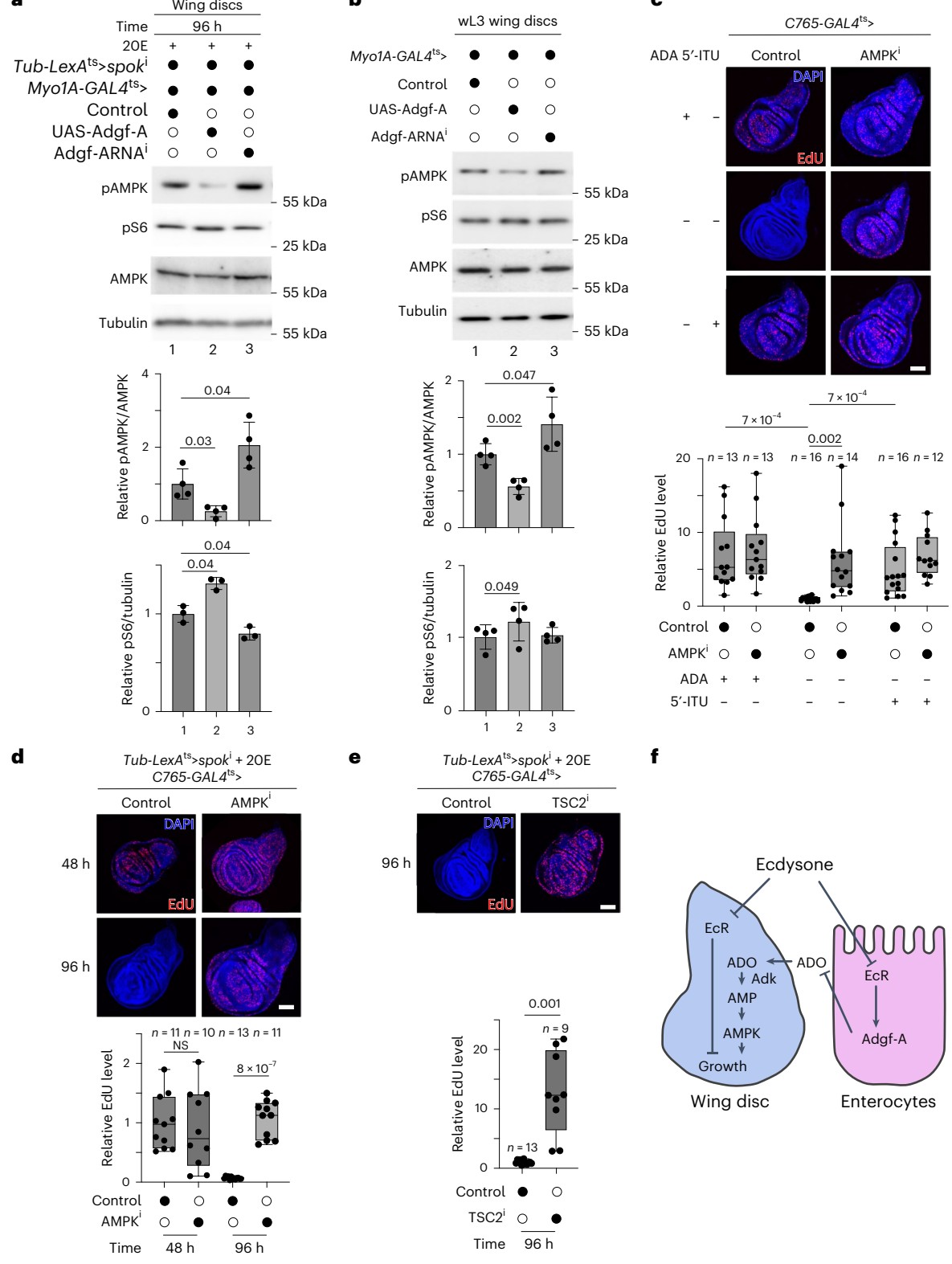

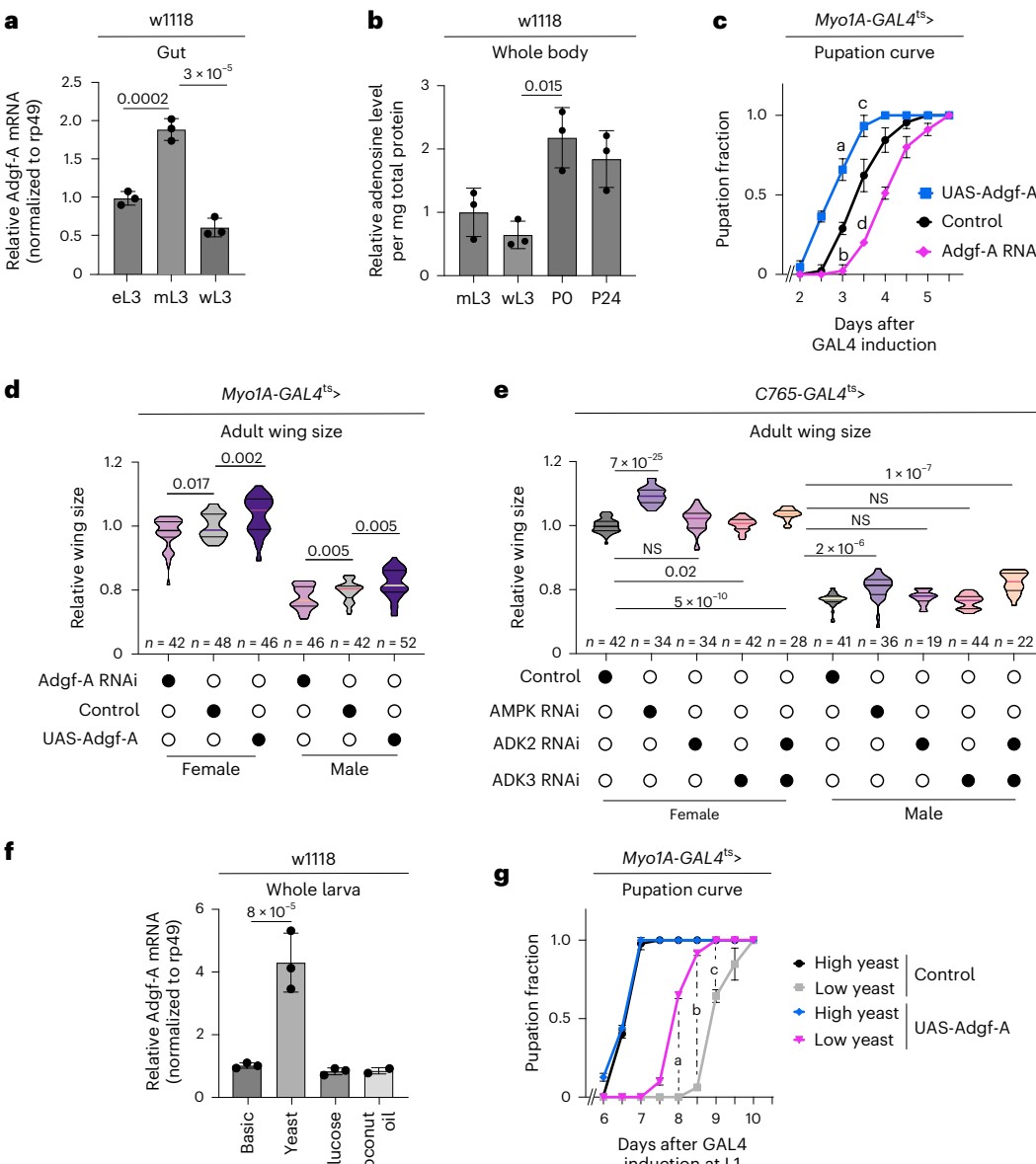

**Fig. 5 | Regulation of wing growth by adenosine via AMPK is regulated in vivo.**
**a**, Adgf-A transcript levels increase in the middle of third instar (L3), quantified by qRT–PCR from whole larvae. $n = 5$ larvae in early L3 (eL3), mid-L3 (mL3) and wandering (wL3) stages. Representative of three biological replicates. Statistical analysis by one-way ANOVA and Dunnett's multiple comparisons test. **b**, Adenosine levels increase in white pupae. Adenosine levels were measured in mL3 and wL3 larvae, white pupae (P0) and pupae 24 h post pupation (P24), normalized to total protein measured by BCA. $n = 3$ whole animals per condition. Statistical analysis by one-way ANOVA and Dunnett's multiple comparisons test. **c**, Circulating adenosine levels regulate developmental timing, assayed by either overexpressing or knocking down *Adgf-A* in intestinal enterocytes to decrease or increase circulating adenosine levels, respectively, and quantifying pupation timing. First instar larvae were collected and seeded at equal density into vials and placed at 18 °C to grow. At early L3, they were shifted to 29 °C to turn on GAL4 activity. $n = 3 \times 30$ animals. Statistical analysis by one-way ANOVA with Dunnett's multiple comparisons test relative to the control genotype, $P = $ (**a**) 0.007, (**b**) 0.002, (**c**) 0.03, (**d**) 0.03. **d**, Adult wings are slightly larger in animals that had low circulating adenosine during L3, generated by overexpressing Adgf-A in intestinal enterocytes, whereas animals with high adenosine have slightly smaller wings. Larvae with indicated genotypes were shifted at early L3 to 29 °C to enable

GAL4 activity. Statistical analysis by unpaired two-tailed *t*-test. **e**, Activity of the adenosine signalling pathway in wings regulates wing size, seen by knocking down *AMPK*, or *Adk2* or *Adk3*. Larvae of indicated genotypes were shifted at early L3 to 29 °C to enable GAL4 activity. Statistical analysis by unpaired two-tailed *t*-test. **f**, Expression of Adgf-A is upregulated upon yeast feeding. Wild-type larvae at early L3 were fed PBS + agar (basic) food supplemented with yeast, glucose or coconut oil for 48 h and Adgf-A mRNA levels were measured by qRT–PCR. $n = 5$ larvae per condition. Representative of three biological replicates. Statistical analysis by unpaired two-tailed *t*-test. **g**, Dietary yeast increases growth rate in part by regulating circulating adenosine levels, seen as a partial rescue in the developmental delay caused by low yeast by reducing circulating adenosine with Adgf-A overexpression. First instar larvae of indicated genotypes were seeded at equal density in food containing high or low yeast at 29 °C to enable GAL4 expression, and then pupation rates were recorded. $n = 3 \times 45$ animals. Significance comparing UAS-Adgf-A animals with control animals by repeated measures two-way ANOVA with Tukey's multiple comparisons test. $P = $ (**a**) $3 \times 10^{-5}$, (**b**) $1 \times 10^{-15}$, (**c**) 0.02. The number of biological replicates and $P$ values are indicated in the graphs; NS, not significant. Bar plots and line plots show mean ± s.d. Violin plots, centre line (median), black lines (first and third quartiles) and the range of data; the distribution is medium smoothed.

The results presented here indicate that *Drosophila* has a circulating, signalling metabolite, adenosine, which actively inhibits tissue growth. This adenosine counteracts insulin, which is also in circulation but promotes tissue growth, whereas adenosine inhibits it. Adenosine levels integrate multiple inputs, as they are regulated by both ecdysone and dietary protein. In conditions of low yeast, adenosine acts as a 'break' because overexpression of Adgf-A causes the animals to grow and develop faster, indicating that they have the metabolic capacity to grow faster than they do. We speculate that biological systems use nutrient sensing pathways in a conservative manner, downregulating cellular metabolism more drastically than necessary based on the available nutrients, to ensure that cells do not encounter a metabolic catastrophe, which could lead to cell death.

We find that ecdysone affects wing growth in two ambivalent ways. On the one hand, it autonomously promotes wing proliferation. On the other hand, it inhibits wing proliferation with a delay via circulating adenosine. Additional work will be required to further dissect the implications of this ambivalent signalling pathway on organ growth. The autonomous branch affects one organ at a time (for example if ecdysone sensitivity is increased in the wing disc, the wing disc will proliferate more, but not other tissues). Thus, autonomous ecdysone signalling might affect the relative proportion of organs. The non-autonomous branch, instead, affects all organs equally (we see that manipulation of circulating adenosine levels either speeds up or slows down the rate of growth of the entire animal). This non-autonomous branch also highlights that organ size control relies on inter-organ communication[55,56]. Finally, due to the time delay in the non-autonomous inhibitory branch, every peak of ecdysone at a larval molt might license organs for a discrete amount of growth. This adds an additional, temporal, aspect to the complexity of how ecdysone affects tissue growth, which was already known to be dose-dependent. Whereas low levels of ecdysone are required for wing growth[22–27], the high levels of ecdysone that induce metamorphosis cause wing discs cells to stop proliferating[17]. Dissecting the underlying logic to this complex wiring may shed light on how organ size and growth rates are regulated.

## Online content

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

## Methods

### *Drosophila* maintenance

A list of fly stocks used in this study are listed in Supplementary Table 1. The sequences of oligonucleotides used for genotyping stocks and recombinant chromosomes are listed in Supplementary Table 2. Detailed genotypes of all animals for all figure panels are indicated in Supplementary Table 3.

Fly stocks were maintained at 25 °C with a 12-h light–dark cycle. Larvae for experiments were raised at 18 °C for 6 days after egg laying to pass the L2/L3 molt, and then shifted to 29 °C to inhibit Gal80[ts] and induce the UAS/GAL4 and LexO/LexA:GAD systems.

For 20E feeding experiment, early L3 larvae were picked manually and transferred into food containing 25 µM 20E. Every 48 h the larvae were transferred into food with fresh 20E. The larvae were kept at 29 °C until the time points indicated in the figures.

For different nutrients conditions, larvae were fed food (PBS + 0.8% agar) mixed with 10% yeast, 10% sucrose or 10% coconut oil. After 2 days, the larvae were used for RT–PCR. For pupation curves of larvae growing on different nutrients conditions, larvae were fed diluted food (20% normal food + 0.8% agar) or diluted food supplemented with 0%, 5% or 10% dry yeast (final). For pupation curves of larvae fed with adenosine, equal numbers of L1 larvae for each genotype were picked and added into food containing 0 mM, 0.5 mM or 1 mM adenosine. For western blots of larvae fed with adenosine diet, equal numbers of L2/L3 molting larvae were picked and added into food containing 0 mM and 1 mM adenosine. After 48 h, the wing discs were dissected and lysed for blotting analysis.

### Explant culture

Third instar wandering larvae were dissected in PBS, and the inverted larvae with attached wing discs were transferred briefly into Shields and Sang M3 insect medium (Sigma, S3652) to wash them. The inverted larvae (6–8 per sample) were then placed in explant medium as described previously[23] (Shields and Sang M3 insect medium containing 100 U ml$^{-1}$ penicillin, 100 µg ml$^{-1}$ streptomycin (Gibco, 15140122), 1.6 nM juvenile hormone (Sigma, 333725), 5 nM ecdysone (Enzo Life Sciences, LKT-E0813-M010) and 10 µg ml$^{-1}$ insulin (Sigma, I0516), with or without 8.3 ng ml$^{-1}$ adenosine deaminase (Roche, 10102105001)) for 3 h. We observed batch-to-batch variation in the amount of adenosine in M3 medium. In cases where the adenosine in the M3 medium was not sufficient to inhibit disc proliferation within 3 h, the medium was also supplemented with 20 µM adenosine. Incubations were performed in explant chambers with constant stirring for oxygenation[23]. For pharmacological treatments, drugs were added to the explant medium with the following final concentrations: adenosine kinase inhibitor 5-Iodotubercidin (10 µM), ENTs inhibitor dipyridamole (30 µM).

### EdU staining

Discs were incubated with EdU (final concentration 25 µM) in Shields and Sang insect M3 medium (for in vivo experiments) or explant culture medium (for explant experiments) while rotating for 1 h at room temperature (RT). Afterwards, discs were fixed in 4% paraformaldehyde for 20 min at RT. Discs were permeabilized in PBS/0.2% Triton (PBT) and blocked in PBT/0.1% BSA (BBT) for 30 min. After removing the solution, EdU Click-IT reaction mix (Invitrogen, C10338; 100 µl per condition) was added and incubated for 35 min at RT. Afterwards, discs were washed in PBT at least three times and nuclei were stained using 4,6-diamidino-2-phenylindole (DAPI) before glycerol mounting medium (160 ml glycerol, 20 ml 10× PBS, 0.8 g n-propyl gallate and 12 ml water) was added for equilibration, and discs were mounted on slides for imaging.

### Quantifications

EdU signals were quantified using a Fiji macro available on GitHub (https://github.com/aurelioteleman/Teleman-Lab/tree/master/ ImageJ%20Macros). This macro quantifies integrated EdU signal normalized to wing disc area, identified via the presence of DAPI/ nuclear stain. Note that this results in values with arbitrary units, and should not be used to compare signal levels between different experiments as the absolute value is influenced by staining and imaging settings. To measure adult wing size, the whole wings were outlined using Fiji (ImageJ) and the area was quantified using the measure function in the software.

### Immunoblotting and antibodies

Twenty wing discs from ten larvae were lysed per sample for immunoblotting. Larvae were dissected in Shields and Sang M3 insect medium and wing discs were transferred into a PCR tube containing the same medium on ice. After spinning them down, the medium was removed and lysis buffer (40 mM HEPES, pH7.5, 10 mM β-glycerophosphate, 10 mM Na-pyrophosphate, 30 mM NaF, 1%Triton X-100, 1× cocktail proteinase inhibitor and 1× phosSTOP) was added immediately. The samples were kept in ice for 10 min, and pipetted up and down several times. After centrifuge at 10,000*g* for 1 min at 4 °C, the supernatant was collected and protein concentration was measured by BCA. Before loading samples on an SDS–PAGE, Laemmli buffer was added to 1× final concentration and the samples were boiled for 5 min at 95 C. Antibodies used are rabbit anti phospho-ribosomal protein S6 1:1,000 dilution[58], rabbit anti phospho-AMPKα (T172) (Cell Signalling, 50081), rabbit anti phospho-S6K (T398) (PhosphoSolutions, p1705-398) 1:1,000 dilution, rabbit anti-AMPKα (Cell Signalling, 2532S) 1:1,000 dilution and rabbit anti-α-tubulin (Cell Signalling, 2125) 1:1,000 dilution. Guinea pig anti-*Drosophila* S6K (1:2,000 dilution) was generated by us and described and validated elsewhere[59].

### Quantitative RT–PCR

Oligonucleotides used for qRT–PCR are listed in Supplementary Table 2. In total, 6–10 wing discs were used for total RNA extraction with TRIZOL (Invitrogen). Reverse transcription was performed with Maxima H Minus Reverse Transcriptase (Thermo Scientific). qPCR was performed using Maxima SYBR Green/Rox (Fermentas), normalized to rp49. The annealing temperature was 60 °C.

### Adenosine quantification

A total of 3 µl of haemolymph from five larvae or three whole animals in each condition were used for analysis with adenosine assay kit (Abcam, ab211094). Larvae were washed with PBS and dried on a clean tissue paper. For quantifying adenosine in haemolymph, five larvae were collected together on a cold glass. The larvae were pinched to break their skin. The haemolymph was collected and immediately mixed with cold assay buffer. For adenosine quantification in whole animals, three larvae or pupae were collected in cold assay buffer and immediately homogenized on ice. After 10 min, the samples were centrifuged at 10,000*g*, 4 °C for 10 min and the supernatant was used for analysis. The assay was then performed as per manufacturer's instructions. The fluorescent signals were read using a SPECTROstar Omega at Ex/Em = 544/585–595 nm with gain at 500. The adenosine levels in whole animals were normalized with total protein amount quantified by BCA assay.

### AMP quantification

Roughly 30 wing discs in each condition were used for analysis using the AMP assay kit (Abcam, ab273275). Roughly 15 larvae were dissected in cold PBS. The wing discs were collected in PCR tubes, and afterward homogenized in cold assay buffer quickly. The samples were kept on ice for 10 min, followed by centrifugation at 10,000*g* for 10 min at 4 °C. The supernatant was collected for analysis as per manufacturer's instructions. The OD$_{570}$ values of all samples were in the linear range, determined by a standard curve. The final values were normalized with total protein amount quantified by BCA assay.

## RNA-seq

*Spok* knockdown and *EcR* knockdown were achieved by crossing *LexO-spok RNAi, Tub-GAL80ts; UAS-EcR RNAi/ST* females with *Myo1A-GAL4; Tub-LexA/ST* males (*EcR*[i]) or *+; Tub-LexA* males (control). To collect experimental animals, flies were allowed to lay eggs in vials for roughly 8 h. Tubes were kept at 18 °C for 6 days to reach the L2/L3 molt. Non-Tubby larvae were picked and transferred to 29 °C incubation for 4 days. The whole intestine from larvae were dissected and used for total RNA extraction with TRIZOL (Invitrogen). The library preparation and sequencing were performed by the DKFZ NGS core facility.

Data analysis was performed on the Galaxy server platform[57]. Before aligning read sequences to the *Drosophila* genome with HISAT2, adaptor sequences were trimmed using Trim Galore! The counts were quantified by featurecounts. The differentially expressed genes were analysed using limma-voom. The genome annotation used was Ensembl (dm6) available at http://ensembl.org. Raw data are deposited at the NCBI Gene Expression Omnibus under accession code GSE284402. Raw counts are provided in Supplementary Table 4.

## Statistics and reproducibility

Statistical analysis was performed using Graphpad Prism v.10. No statistical method was used to predetermine sample size. No data were excluded from the analyses. The experiments were randomized. The investigators were not blinded to allocation during experiments and outcome assessment. For each dataset, we first used the Shapiro–Wilk test to check for data normality. For data that followed a normal distribution, we used a two-tailed *t*-test for pairwise comparisons, and ANOVA with Dunnett's multiple comparison correction for multi-group comparisons. For data that did not follow normality, we used a two-tailed Mann–Whitney *U*-test for pairwise comparisons and a Kruskal–Wallis test with Dunn's post hoc correction for multi-group comparisons. The tests used are indicated in the figure legends. *P* values < 0.05 were considered as significant and are shown in the figures. Figures were prepared with Affinity Photo.

## Reporting summary

Further information on research design is available in the Nature Portfolio Reporting Summary linked to this article.

## Data availability

RNA-seq data are deposited at the NCBI Gene Expression Omnibus under accession code GSE284402 and raw counts are provided in Supplementary Table 4. All other data are available in the main text or the supplementary materials. Source data are provided with this paper.

## Code availability

EdU signals were quantified using a Fiji macro available on GitHub (https://github.com/aurelioteleman/Teleman-Lab/tree/master/ImageJ%20Macros).

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

## Acknowledgements

We thank the DKFZ Genomics Core for next-generation sequencing, and the Boutros laboratory, Dolezal laboratory and Rewitz laboratory for fly stocks.

## Author contributions

Y.Z. and A.A.T. conceived of the project. Y.Z. and K.S. performed the experiments. Y.Z. and A.A.T. analysed the data. Y.Z. and A.A.T. wrote the paper.

## Funding

## Competing interests

The authors declare no competing interests.

## Additional information

**Extended data** is available for this paper at https://doi.org/10.1038/s41556-025-01764-0.

**Correspondence and requests for materials** should be addressed to Aurelio A. Teleman.

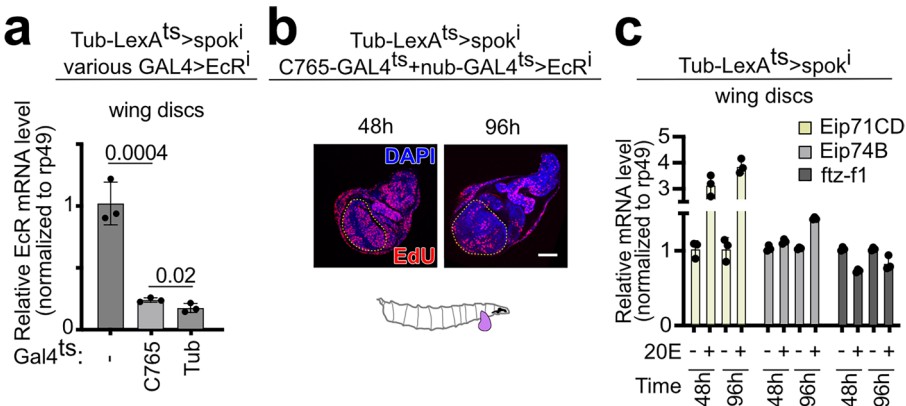

**Extended Data Fig. 1 | Wing-specific knockdown of EcR enables wing discs to proliferate.** (**a**) *EcR* knockdown either ubiquitously with *Tub-GAL4* or specifically in the wing disc with *C765-GAL4* reduces *EcR* mRNA levels in the wing disc, assayed by QRT–PCR. Ten wing discs were user per condition. Representative of three biological replicates. Statistical analysis by unpaired two-tailed t-test. (**b**) Wing-specific knockdown of *EcR* in animals lacking ecdysone enables wing discs to proliferate without stopping, even if EcR levels are knocked down strongly using a combination of *C765-GAL4* and *nub-GAL4*. Lack of ecdysone is induced by knockdown of spookier (*spok*^i) with a LexA driver. Representative images from 10 or 12 wing discs at 48 h or 96 h, respectively. Pouch is circled with a yellow dashed line. (**c**) Ecdysone signalling is still active in wing discs at 96 h of ecdysone feeding to animals lacking endogenous ecdysone production (when wing discs stop proliferating). Larvae expressing *spok* RNAi (*spok*^i) were fed with 25 µM 20E or with vehicle for 48 h or 96 h (refreshing the treatment every 48 h), and RNA was extracted from ten wing discs per condition. Expression of ecdysone targets was measured by QRT–PCR. Note that ftz-f1 is repressed by ecdysone signalling. Representative of three biological replicates. All Panels: p-values are indicated in the graphs; ns: not significant. Bar plots: mean ± std. dev. Box plots: centre line (median), box limits (first and third quartiles) and whisker (outer data points). Scale bars: 100 µm. Source numerical data are available in.

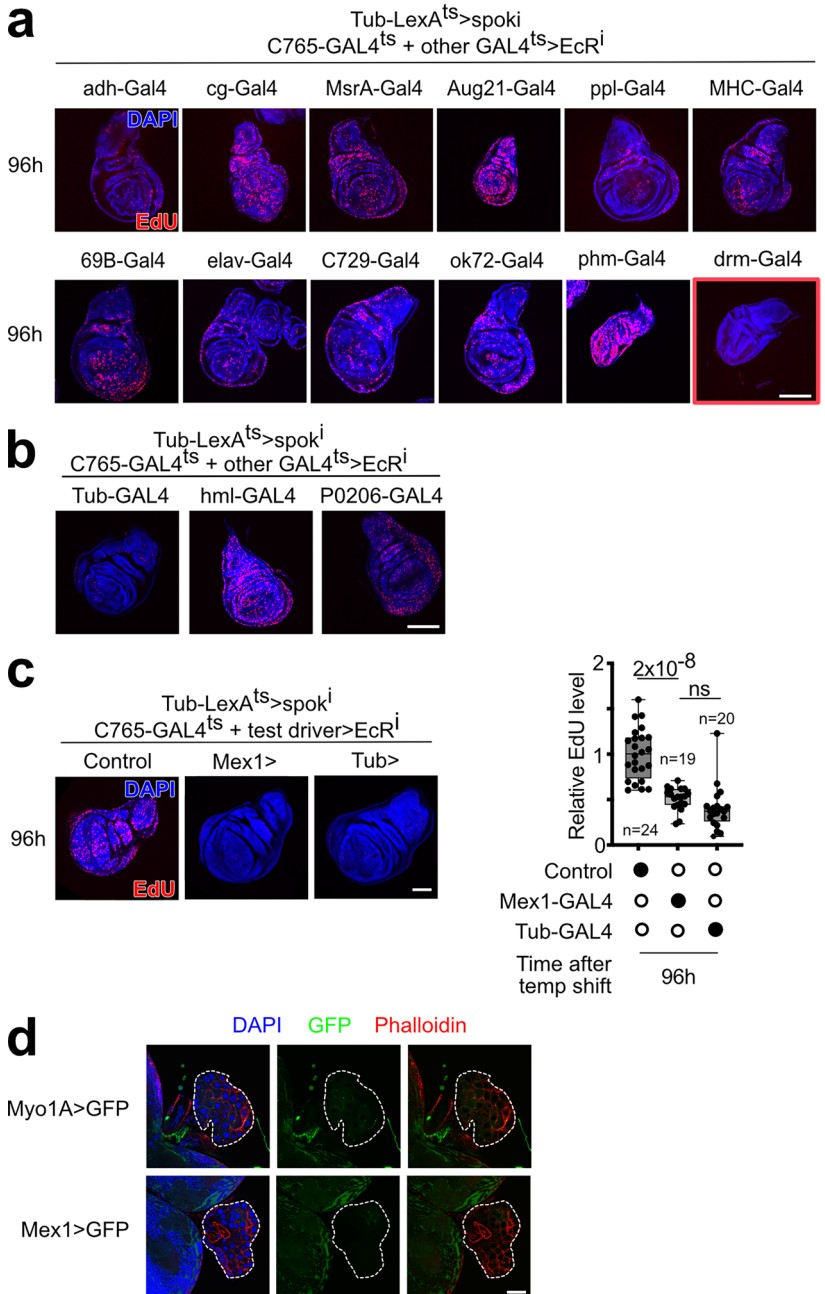

**Extended Data Fig. 2 | Screen to identify the other tissue through which ecdysone acts to inhibit wing disc proliferation. (a, b)** Screen of different GAL4 drivers identifies the intestine as the target organ through which ecdysone inhibits wing disc proliferation non-autonomously. All larvae lack endogenous ecdysone production (spok RNAi) but have a wing disc specific knockdown of *EcR* which enables wing discs to proliferate indefinitely. Into this genetic background, additional GAL4 drivers were added to knockdown *EcR* in other tissues, and wing proliferation was assayed 96 h after inducing GAL4 activity with a temperature shift at the beginning of L3. Representative images are shown, from at least 7 wing discs assayed in each condition. Scale bars: 100 µm. **(c)** Knockdown of *EcR* in enterocytes with *Mex1-GAL4* is sufficient to cause termination of wing disc proliferation at 96 h. Larvae lacking endogenous ecdysone synthesis due to spok[i], and with wing disc knockdown of *EcR* using *C765-GAL4* which enables wing discs to proliferate indefinitely, were combined with GAL4 drivers targeting either enterocytes (*Mex1-GAL4*) or all tissues (*Tub-GAL4*). (Left) Representative images. (Right) Quantification of EdU at 96 h after RNAi induction. Statistical analysis by one-way ANOVA and Dunnett's multiple comparisons test. Scale bar: 50 µm. **(d)** *Myo1A-GAL4* and *Mex1-GAL4* are almost not expressed in the ring gland. *Myo1A-GAL4* and *Mex1-GAL4* were crossed to UAS-GFP and imaged. The laser intensity was increased so as to give some background signal in the brain. Expression in the ring glad, outlined with a dashed trace, is even lower. Scale bar: 25 µm. All Panels: The number of biological replicates and p-values are indicated in the graphs; ns: not significant. Box plots: centre line (median), box limits (first and third quartiles) and whisker (outer data points). Source numerical data are available in.

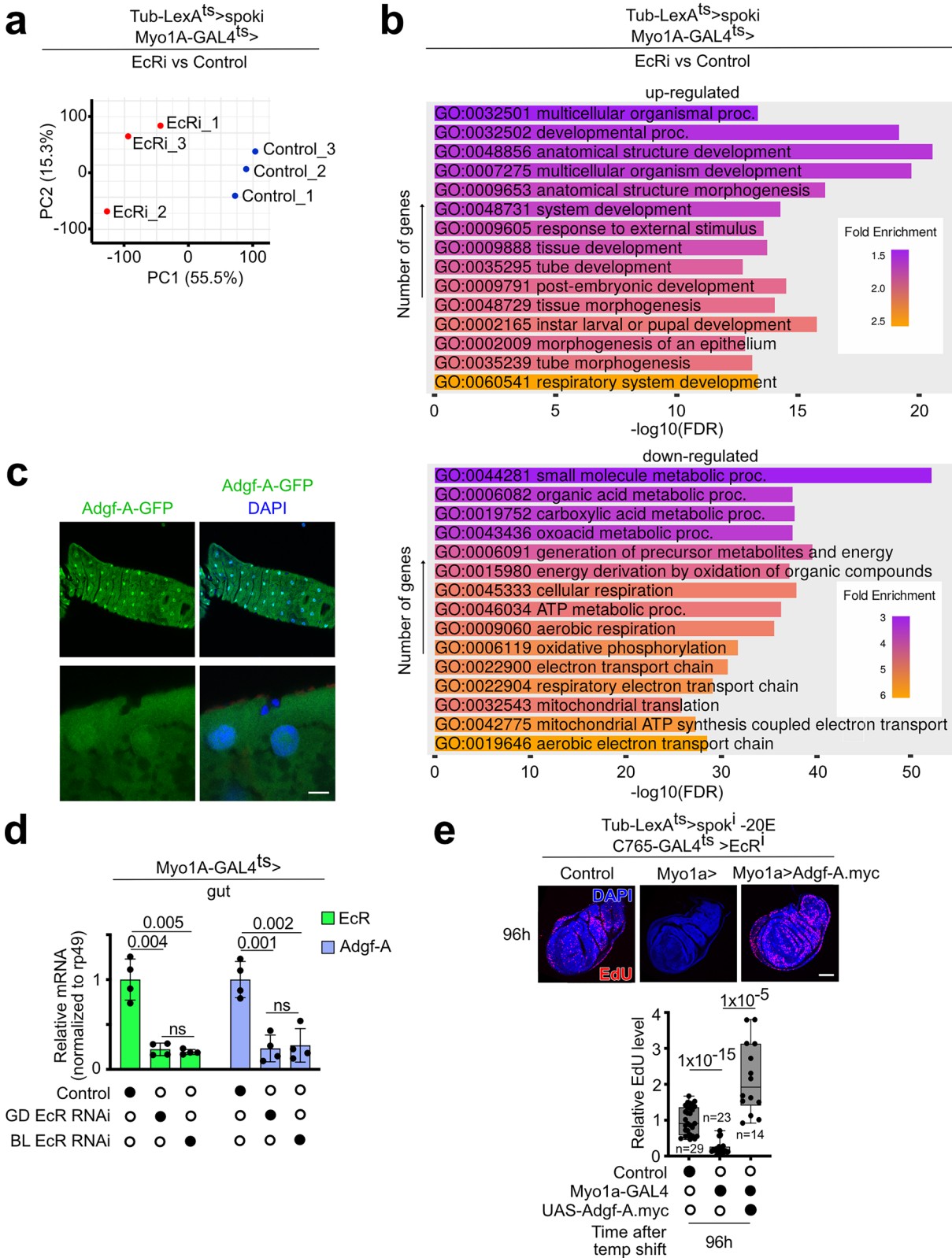

**Extended Data Fig. 3 | See next page for caption.**

**Extended Data Fig. 3 | Analysis of RNA-seq on intestines from animals with or without EcR knockdown in intestinal enterocytes.** (a) Principal component analysis (PCA) shows that the samples cluster by genotype. RNA was extracted from intestines of larvae that have no endogenous ecdysone synthesis (spoki), either with or without knockdown of *EcR* in enterocytes driven by *Myo1A-GAL4*. All 3 replicates for each genotype are shown. Singular Value Decomposition (SVD) with imputation is used to calculate principal components. (**b**) Gene Ontology enrichment analysis for Biological Process terms amongst either up-regulated (top) or downregulated (bottom) genes in the *EcR*-RNAi condition compared to control. FDR: false discovery rate. (**c**) Adgf-A is expressed in all enterocytes of the gut, seen via a GFP insertion into the endogenous Adgf-A genomic locus 39. (Upper) Representative image of a midgut. (Lower) Small cells such as intestinal stem cells have reduced expression of Adgf-A. (**d**) Knockdown of *EcR* using two independent RNAi lines driven by *Myo1A-GAL4* in the gut leads to reduced expression of Adgf-A in the gut, quantified by QRT–PCR. GD *EcR* RNAi = VDRC line # 37059, BL *EcR* RNAi = Bloomington line #9327 (*P{w[+mC]=UAS-EcR. RNAi.C}104*). Statistical analysis by unpaired two-tailed t-test. n = 4 ×10 discs. (**e**) Expression of Adgf-A rescues the inhibition of wing proliferation caused by activation of EcR target genes in intestinal enterocytes. Statistical analysis by one-way ANOVA and Dunnett's multiple comparisons test. All Panels: The number of biological replicates and p-values are indicated in the graphs; ns: not significant. Bar plots: mean ± std. dev. Box plots: centre line (median), box limits (first and third quartiles) and whisker (outer data points). Scale bars: 5 µm (**c**), 100 µm (**e**). Source numerical data are available in.

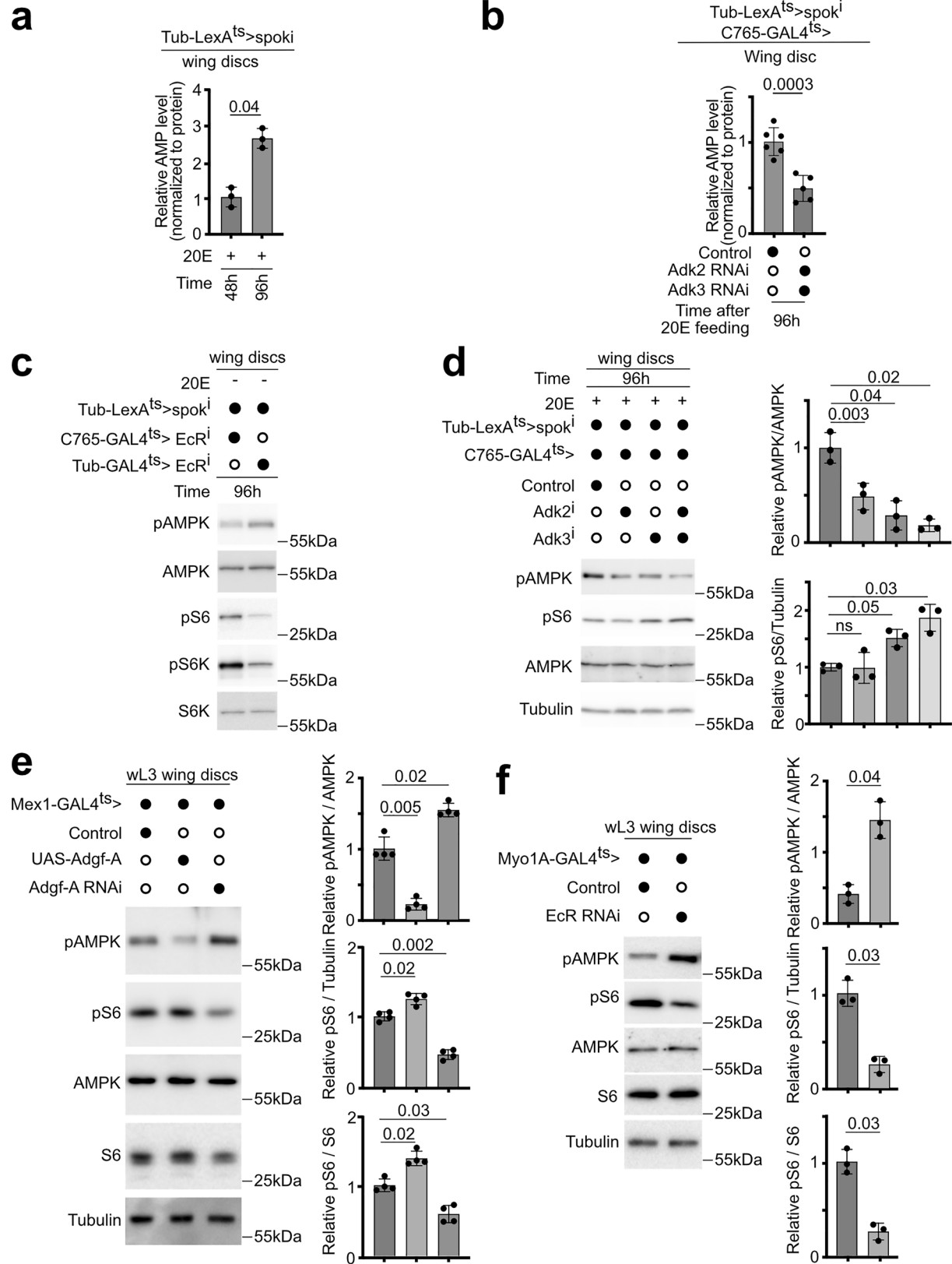

**Extended Data Fig. 4 | See next page for caption.**

**Extended Data Fig. 4 | Ecdysone signalling increases AMPK activity in wing discs via AMP.** (**a**) AMP levels in wing discs increase at 96 h after 20E feeding due to Adk activity. AMP was measured in lysates from 20 wing discs from spoki larvae fed with 25 μM 20E for 48 h or 96 h (**a**) or in similar discs containing simultaneous knockdown of *Adk2* and *Adk3* (**b**). AMP levels are normalized to total protein amount. (**a**) n = 3 ×20 wings. (**b**) 6 ×20 wings for control and 5 ×20 wings for ADKs KD. Statistical analysis by unpaired two-tailed t-test. (**c**) Knockdown of *EcR* in tissues other than the wing leads to increased AMPK activity and reduced mTORC1 activity in wing discs. Wing discs from animals lacking endogenous ecdysone synthesis with an *EcR* knockdown either specifically in the wing with *C765-GAL4*, or ubiquitously with *Tub-GAL4*, were lysed 96 h after induction of GAL4 activity and immunoblotted. n = 20 discs per sample. Representative of two biological replicates. (**d**) Combined knockdown of adenosine kinases *Adk2* and *Adk3* in wing discs reduces AMPK activation and increases mTORC1 activity, assayed via immunoblotting of wing disc lysates. All animals lack endogenous

ecdysone biosynthesis (spoki) and are fed 25 μM for 96 h. (top) Representative immunoblot. (bottom) Quantification of 3 biological replicates, 20 wing discs per sample. Statistical analysis by paired two-tailed t-test. (**e**) Circulating adenosine levels regulate wing AMPK and mTORC1 activity, assayed by either overexpressing or knocking down *Adgf-A* in enterocytes (with *Mex1-GAL4*) in animals that are otherwise wild-type. Activities are assayed by immunoblotting wing disc lysates. (Left) Representative immunoblot. (Right) Quantification of 4 replicates, 20 wing discs per sample. Statistical analysis by paired two-tailed t-test. (**f**) Activation of EcR target genes in the gut by knocking down *EcR* with *Myo1A-GAL4* leads to AMPK activation and mTORC1 inhibition in wing discs. (Left) representative example of wing disc lysates. (Right) Quantification of 3 biological replicates, 20 wing discs per sample. Statistical analysis by paired two-tailed t-test. All panels: p-values are indicated in the graphs; ns: not significant. Bar plots: mean ± std. dev. Source numerical data and unprocessed blots are available in.

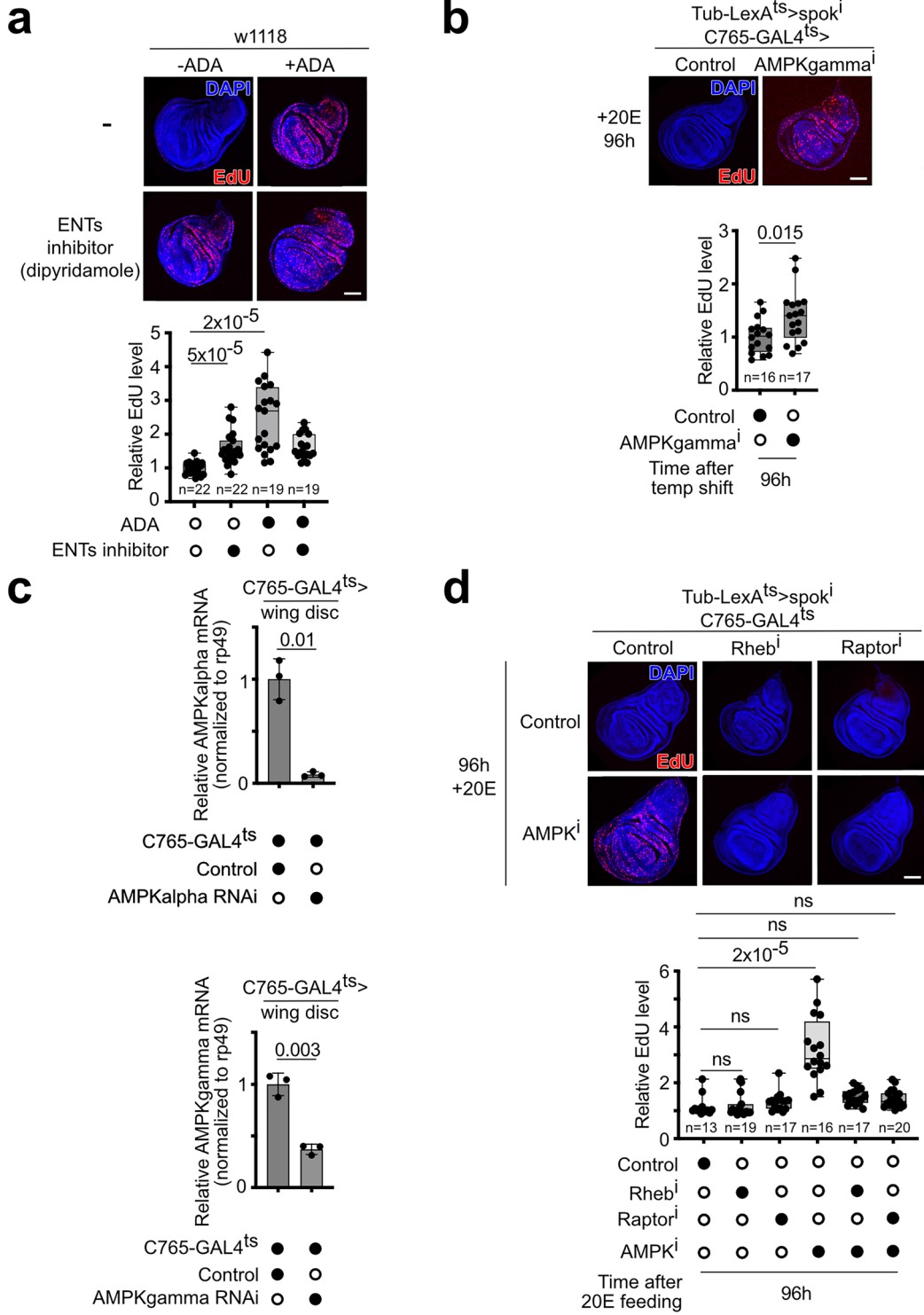

**Extended Data Fig. 5 | Ecdysone signalling at 96 h inhibits wing disc proliferation via activation of AMPK.** (**a**) Adenosine in explant culture needs to enter into cells to inhibit wing disc proliferation, seen by pharmacologically blocking Equilibrative Nucleoside Transporters (ENTs) with the inhibitor dipyridamole (30 μM). Both the explants and the ENT inhibition were done for 3 h. (Top) representative images. (Bottom) Quantification of EdU. Statistical analysis by one-way ANOVA and Dunnett's multiple comparisons test. (**b**) Knockdown of the gamma subunit of AMPK in wing discs of animals lacking endogenous ecdysone biosynthesis (spoki) but fed 25 μM 20E enables wing discs to continue proliferating at 96 h, showing that AMPK activation causes termination of wing proliferation. (Top) Representative images. (Bottom) Quantification of EdU. Statistical analysis by unpaired two-tailed

t-test. (**c**) Knockdown of the gamma subunit of AMPK is less efficient than the knockdown of the alpha subunit, quantified by QRT–PCR from wing disc RNA. Statistical analysis by unpaired two-tailed t-test. n = 3 ×10 discs. (**d**) Inhibition of mTORC1, by knocking down either *Rheb* or *Raptor*, reverses the proliferation enabled by *AMPK* knockdown in animals lacking endogenous ecdysone biosynthesis (spoki) but fed 25 μM 20E for 96 h. (Top) Representative images. (Bottom) Quantification of EdU. Statistical analysis by one-way ANOVA and Dunnett's multiple comparisons test. All panels: The number of biological replicates and p-values are indicated in the graphs; ns: not significant. Bar plots: mean ± std. dev. Box plots: centre line (median), box limits (first and third quartiles) and whisker (outer data points). Scale bars: 100 μm. Source numerical data are available in.

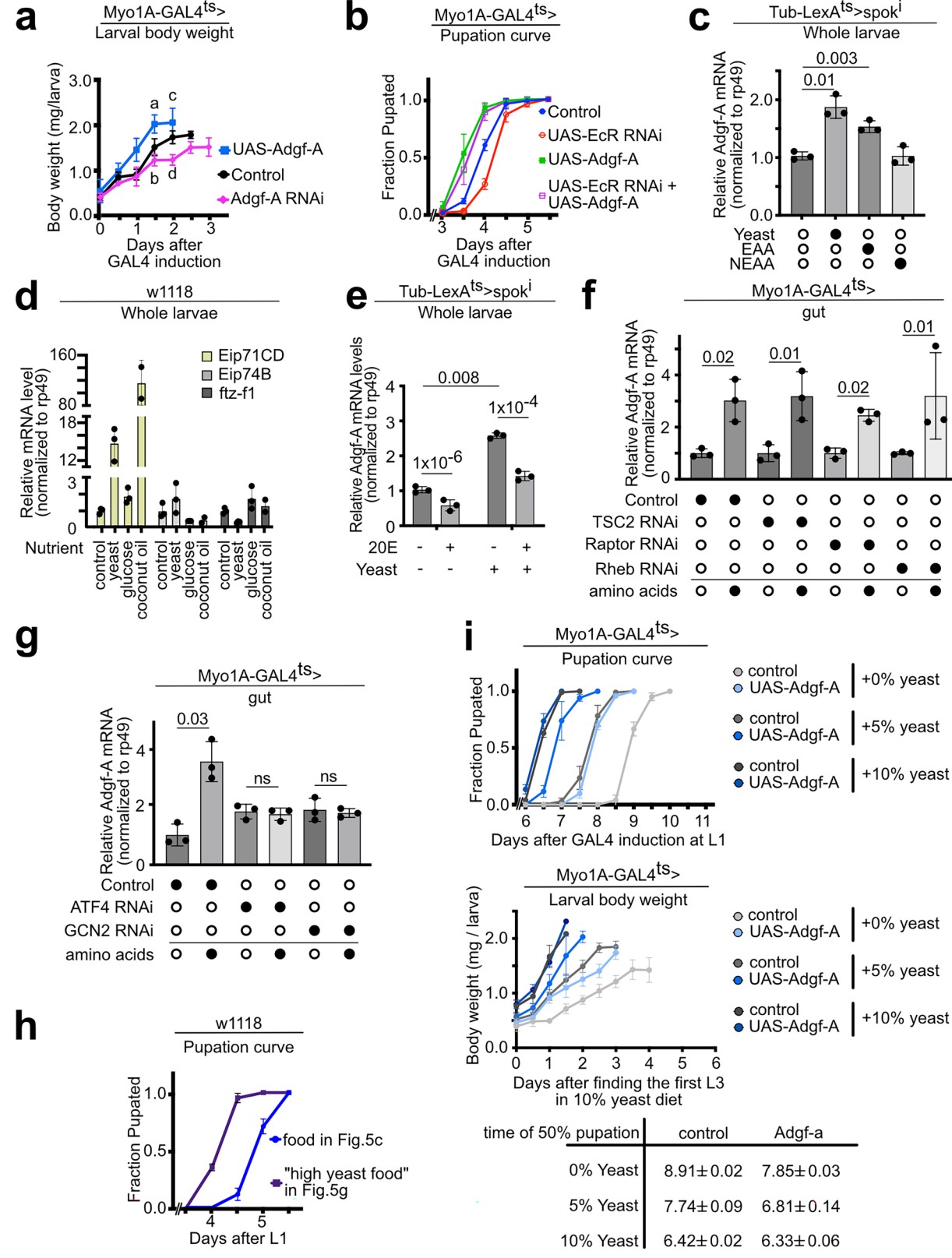

Extended Data Fig. 6 | See next page for caption.

**Extended Data Fig. 6 | Dietary yeast induces Adgf-A expression independently of ecdysone signalling.** (**a**) Circulating adenosine regulates larval growth rate, assayed as total body weight. Adgf-A was overexpressed or knocked down in intestinal enterocytes to decrease or increase circulating adenosine levels, respectively. First instar larvae were collected and seeded at equal density into vials and placed at 18 °C to grow. At early L3, batches of 20 animals were shifted to 29 C to turn on GAL4 activity, and body weight was measured twice per day. n = 5 ×20 animals. Statistical analysis by one-way ANOVA with Dunnett's multiple comparisons test relative to the control genotype, p = (**a**) 0.001, (**b**) 0.04, (**c**) 0.049, (**d**) 0.005 (**b**) Adgf-A expression rescues the pupation delay caused by *EcR* knockdown in enterocytes. First instar larvae were collected and seeded at equal density into vials and placed at 18 °C to grow. At L1, batches of 30 animals were shifted to 29 C to turn on GAL4 activity. n = 5 ×30 animals. Statistical analysis by two-way ANOVA test, p = 0.046 means Adgf-a has different effect at all values of EcR background (**c**) Dietary essential amino acids increase Adgf-A expression. Early L3 spoki larvae were placed onto different diets containing PBS + 1%agar supplemented with nothing (control), 10% dry yeast, 5X (final) MEM essential amino acids (EAA), or 10X (final) non-essential amino acids (NEAA) at 29 C for 2 days to remove endogenous ecdysone. Whole larvae were used for analysis. n = 3 x biological replicates. Statistical analysis by unpaired two-tailed t-test. (**d**) Yeast supplementation does not cause a decrease in ecdysone signalling, assayed by QRT–PCR on three ecdysone target genes: Eip71CD and Eip74B, which are induced by ecdysone signalling, and ftz-f1 which is repressed. Wild-type larvae at early L3 were fed with yeast, glucose or coconut oil in PBS+agar (basic) food for 48 h. Five larvae were used for analysis per condition. Representative of three biological replicates. (**e**) *Adgf-A* transcription is induced by dietary yeast also in larvae lacking ecdysone signalling due to knockdown of endogenous ecdysone synthesis (*spok^i*). *Spok^i* larvae were fed a diet with or without yeast combined with or without 25 µM 20E in PBS+agar food for 48 h. Five larvae were

used per condition. Adgf-A mRNA was measured by QRT–PCR. Representative of three biological replicates. Statistical analysis by one-way ANOVA and Dunnett's multiple comparisons test. (**f**) The induction of gut Adgf-A expression in response to dietary essential amino acids is not blunted if mTORC1 activity in intestinal enterocytes is either maximally induced by knockdown of *TSC2* or maximally repressed by knockdown of either *Raptor* or *Rheb*. *Adgf-A* expression was quantified by QRT–PCR on RNA extracted from intestines of animals with the indicated genotypes. n = 3 biological replicates. Statistical analysis by one-way ANOVA and Dunnett's multiple comparisons test. (**g**) The induction of gut Adgf-A expression in response to dietary essential amino acids is blunted if either *GCN2* or the downstream transcription factor *ATF4* is knocked down in enterocytes. Adgf-A expression was quantified by QRT–PCR on RNA extracted from intestines of animals with the indicated genotypes. n = 3 biological replicates. Statistical analysis by one-way ANOVA and Dunnett's multiple comparisons test. (**h**) The "high yeast" food used in Fig. 5g and Extended Data Fig. 6i, supplemented with 10% dry yeast, is more rich than our standard fly food used in all other experiments in this paper, including in Fig. 5c, assayed via the rate of pupation of control animals. n = 3 ×30 animals. (**i**) Dietary yeast affects growth rate in part via regulation of adenosine levels. Pupation curves of control or Adgf-A overexpressing animals on food supplemented with three different yeast concentrations. First instar larvae of indicated genotypes were seeded at equal density in food supplemented with 10%, 5% or 0% dry yeast at 29 °C to enable GAL4 expression, and then larval body weight (bottom) and pupation rate (top) were assayed twice per day. The timepoints (in days) of 50% pupation, which were used for the 2-way ANOVA, are indicated in the table on the bottom (derived from non-linear regression). For the pupation curve, 3 ×35 animals were used. For body weight quantification, 5 ×20 animals were used. All panels: p-values are indicated in the graphs. Bar plots and line plots show mean ± std. dev. Source numerical data are available in.

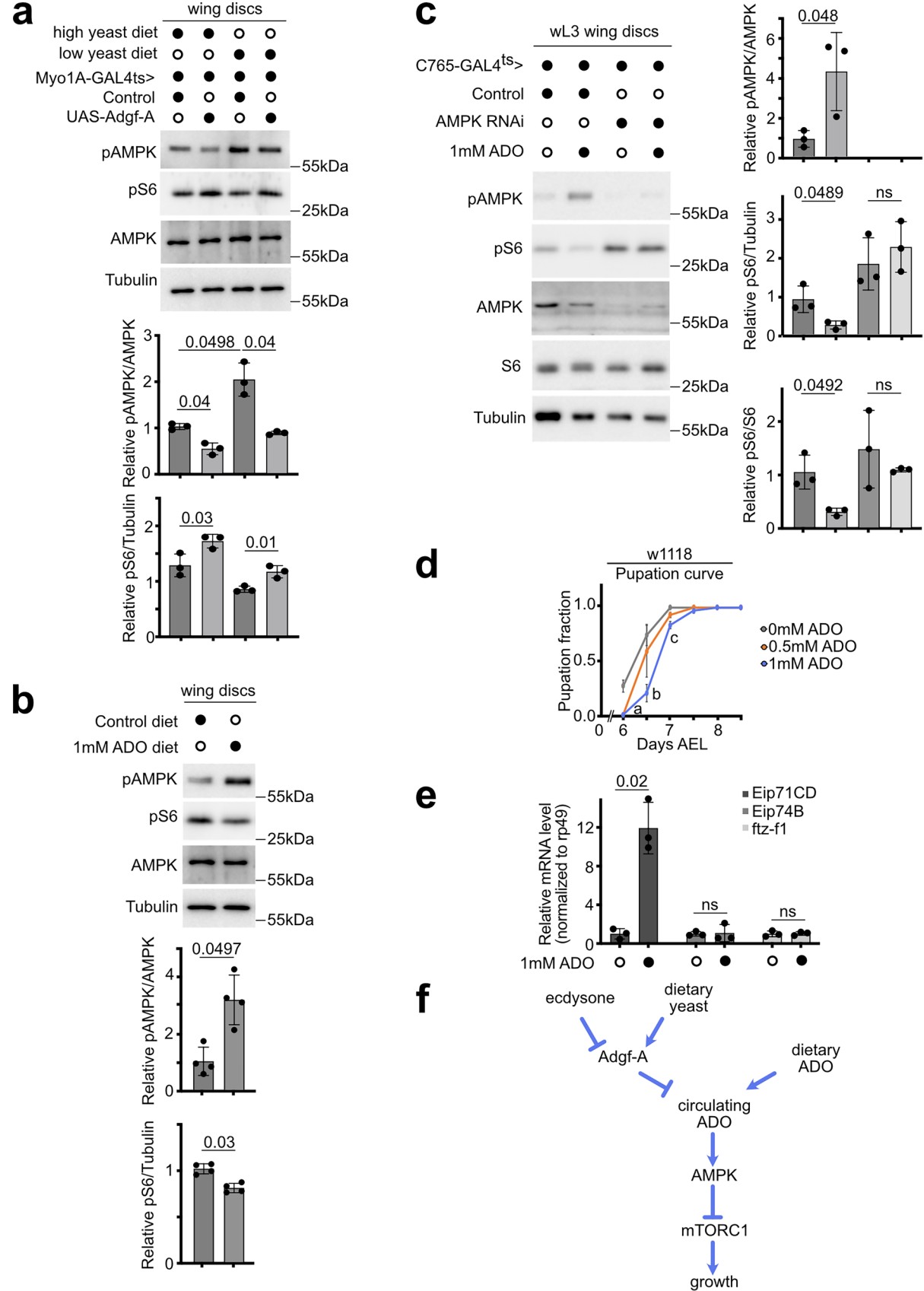

**Extended Data Fig. 7 | See next page for caption.**

**Extended Data Fig. 7 | Adenosine regulates wing disc AMPK activity, mTORC1 activity, and growth.** (**a**) Dietary yeast regulates AMPK and mTORC1 activity in part via Adgf-A and adenosine. Control or Adgf-A overexpressing larvae were placed on food containing either high or low yeast concentrations for 48 h and mTORC1 and AMPK activity was then assayed by immunoblotting wing disc lysates. (Top) Representative immunoblot. (Bottom) Quantification of 3 biological replicates, 20 wing discs per sample. Statistical analysis by paired two-tailed t-test. (**b**) Dietary adenosine regulates wing AMPK and mTORC1 activity, assayed by feeding early L3 larvae either normal food (control diet) or normal food supplemented with 1 mM adenosine for 48 h. Activities are assayed by immunoblotting wing disc lysates. (Top) Representative immunoblot. (Bottom) Quantification of 4 replicates, 20 wing discs per sample. Statistical analysis by paired two-tailed t-test. (**c**) Dietary adenosine inhibits wing disc mTORC1 via AMPK, seen by a loss of mTORC1 regulation when *AMPK* is knocked down in the wing disc. Immunoblots on wing disc lysates from larvae of indicated genotypes fed with or without 1 mM adenosine for 48 h. (Left) representative immunoblot. (Right) quantification of 3 replicates, 20 wing discs per sample.

Statistical analysis by paired two-tailed t-test. (**d**) Dietary adenosine regulates developmental timing, assayed by feeding larvae with either normal food, or normal food supplemented with adenosine, and quantifying pupation timing. First instar w1118 larvae were collected and seeded at equal density into vials and placed at 25 °C to grow. n = 3 ×30 animal. Significance of food supplemented with 1 mM adenosine relative to normal food is shown p = (a)0.02, (b)0.002; (c)0.01. Statistical analysis by unpaired two-tailed t-test. (**e**) Dietary adenosine does not directly regulate ecdysone signalling. Quantitative RT–PCR on RNA from whole larvae fed with or without 1 mM adenosine for 24 h. Eip71CD and Eip74B are induced by ecdysone signalling, whereas ftz-f1 is repressed by ecdysone signalling. n = 3 biological replicates. Statistical analysis by unpaired two-tailed t-test. (**f**) Schematic diagram illustrating that circulating adenosine integrates both dietary and hormonal inputs to regulate AMPK, mTORC1 and organismal growth rate. All panels: p-values are indicated in the graphs. Bar plots and line plots show mean ± std. dev. Source numerical data and unprocessed blots are available in.

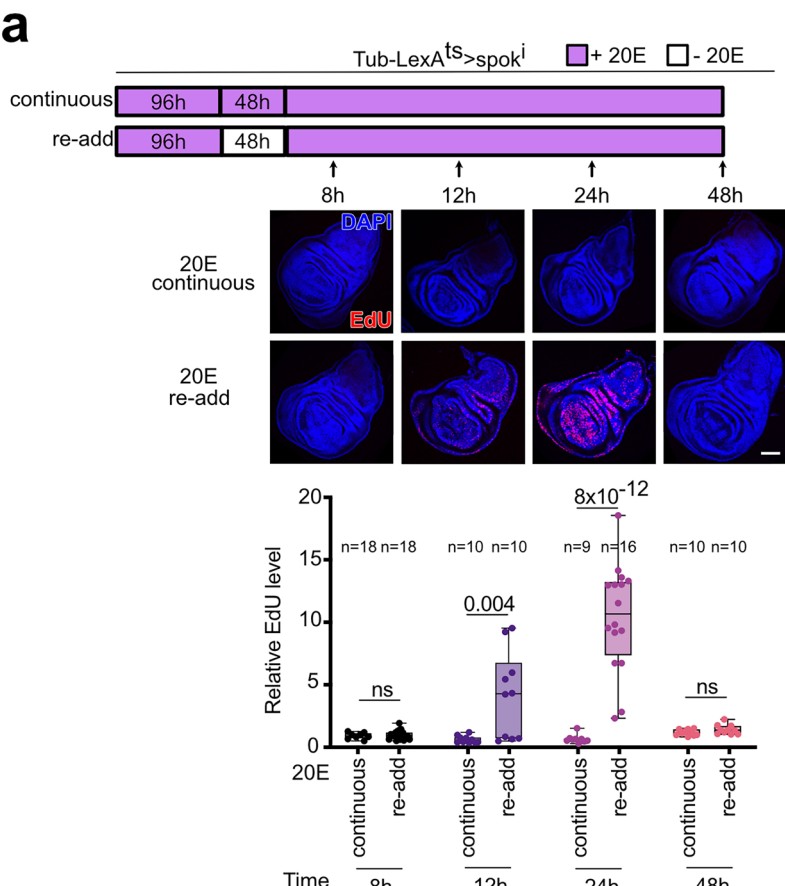

**Extended Data Fig. 8 | Every pulse of ecdysone causes a transient induction of wing proliferation.** (**a**) Removal of ecdysone for 48 h enables a subsequent pulse of ecdysone to again promote proliferation of wing discs for a transient period. (top) Scheme illustrating the two ecdysone feeding conditions. "continuous": larvae were fed with 25 μM 20E continuously; "re-add": larvae were fed with 25 μM 20E for 96 h, then larvae were transferred to food without 20E for 48 h, and finally larvae were transferred back into food containing 25 μM 20E. At 8 h, 12 h, 24 h and 48 h after this final transfer back into 20E-containing food (and the same timepoints in the 'continuous' condition') the wing discs were stained with EdU to assay proliferation. (middle) Representative images. (bottom) Quantification of EdU. Statistical analysis by two-way ANOVA and Sidak multiple comparisons test. p-values are indicated in the graphs; ns: not significant. Box plots: centre line (median), box limits (first and third quartiles) and whisker (outer data points). Source numerical data are available in.

# Reporting Summary

## Statistics

For all statistical analyses, confirm that the following items are present in the figure legend, table legend, main text, or Methods section.

| n/a | Confirmed | |
|---|---|---|
| ☐ | ☒ | The exact sample size (*n*) for each experimental group/condition, given as a discrete number and unit of measurement |
| ☐ | ☒ | A statement on whether measurements were taken from distinct samples or whether the same sample was measured repeatedly |
| ☐ | ☒ | The statistical test(s) used AND whether they are one- or two-sided<br>*Only common tests should be described solely by name; describe more complex techniques in the Methods section.* |
| ☒ | ☐ | A description of all covariates tested |
| ☐ | ☒ | A description of any assumptions or corrections, such as tests of normality and adjustment for multiple comparisons |
| ☐ | ☒ | A full description of the statistical parameters including central tendency (e.g. means) or other basic estimates (e.g. regression coefficient) AND variation (e.g. standard deviation) or associated estimates of uncertainty (e.g. confidence intervals) |
| ☐ | ☒ | For null hypothesis testing, the test statistic (e.g. *F*, *t*, *r*) with confidence intervals, effect sizes, degrees of freedom and *P* value noted<br>*Give P values as exact values whenever suitable.* |
| ☒ | ☐ | For Bayesian analysis, information on the choice of priors and Markov chain Monte Carlo settings |
| ☒ | ☐ | For hierarchical and complex designs, identification of the appropriate level for tests and full reporting of outcomes |
| ☒ | ☐ | Estimates of effect sizes (e.g. Cohen's *d*, Pearson's *r*), indicating how they were calculated |

*Our web collection on statistics for biologists contains articles on many of the points above.*

## Software and code

Policy information about availability of computer code

| Data collection | Leica confocal microscope SP8 with software Leica Application Suite X (LAS X) version 3.5.2.18963 was used. |
|---|---|
| Data analysis | All software is publicly or commercially available:<br>Images analysis: ImageJ/FIJI v 2.14.0.<br>RNA-seq analysis: the galaxy server platform (https://usegalaxy.org)<br>GO enrichment analysis: the online tool http://www.webgestalt.org.<br>Figure preparation: Affinity Photo v1.<br>Data analysis: Microsoft Excel v16 or Graphpad Prism v10.5 |

For manuscripts utilizing custom algorithms or software that are central to the research but not yet described in published literature, software must be made available to editors and reviewers. We strongly encourage code deposition in a community repository (e.g. GitHub). See the Nature Portfolio guidelines for submitting code & software for further information.

## Data

Policy information about availability of data

All manuscripts must include a data availability statement. This statement should provide the following information, where applicable:
- Accession codes, unique identifiers, or web links for publicly available datasets
- A description of any restrictions on data availability
- For clinical datasets or third party data, please ensure that the statement adheres to our policy

RNA-seq data are deposited at NCBI with accession GSE284402 and raw counts are provided in Suppl. Table 4. All other data are available in the main text or the supplementary materials.

## Research involving human participants, their data, or biological material

Policy information about studies with human participants or human data. See also policy information about sex, gender (identity/presentation), and sexual orientation and race, ethnicity and racism.

| Reporting on sex and gender | Not applicable. |
|---|---|
| Reporting on race, ethnicity, or other socially relevant groupings | Not applicable. |
| Population characteristics | Not applicable. |
| Recruitment | Not applicable. |
| Ethics oversight | Not applicable. |

Note that full information on the approval of the study protocol must also be provided in the manuscript.

# Field-specific reporting

Please select the one below that is the best fit for your research. If you are not sure, read the appropriate sections before making your selection.

☒ Life sciences          ☐ Behavioural & social sciences          ☐ Ecological, evolutionary & environmental sciences

For a reference copy of the document with all sections, see nature.com/documents/nr-reporting-summary-flat.pdf

# Life sciences study design

All studies must disclose on these points even when the disclosure is negative.

| Sample size | No sample size calculations were performed. Sample sizes were determined based on practical considerations. Based on these considerations, sample sizes were set at >8 wing discs per genotype/condition, which allowed modest standard deviations and  significant p values to be obtained. |
|---|---|
| Data exclusions | No data were excluded. |
| Replication | The number of replicates is indicated in the figure legends. |
| Randomization | All animals were randomly alloted. |
| Blinding | The experiments were not blinded because they each contained an internal control and because blinding is not usual in this field. |

# Reporting for specific materials, systems and methods

We require information from authors about some types of materials, experimental systems and methods used in many studies. Here, indicate whether each material, system or method listed is relevant to your study. If you are not sure if a list item applies to your research, read the appropriate section before selecting a response.

## Materials & experimental systems

| n/a | Involved in the study |
|-----|----------------------|
| ☐ | ☒ Antibodies |
| ☒ | ☐ Eukaryotic cell lines |
| ☒ | ☐ Palaeontology and archaeology |
| ☐ | ☒ Animals and other organisms |
| ☒ | ☐ Clinical data |
| ☒ | ☐ Dual use research of concern |
| ☒ | ☐ Plants |

## Methods

| n/a | Involved in the study |
|-----|----------------------|
| ☒ | ☐ ChIP-seq |
| ☒ | ☐ Flow cytometry |
| ☒ | ☐ MRI-based neuroimaging |

# Antibodies

| | |
|---|---|
| Antibodies used | rabbit anti phospho-AMPKa(T172) (Cell signaling 50081), rabbit anti phospho-S6K (T398) (PhosphoSolutions p1705-398) 1:1000, rabbit anti-AMPKa (Cell signaling 2532S) 1:1000, rabbit anti-tubulin (Cell signaling 2125) 1:1000, Guinea pig anti Drosophila S6K (1:2000), rabbit anti phospho-ribosomal protein S6 1:1000 |
| Validation | rabbit anti phospho-ribosomal protein S6 1:1000 was validated in the publication: PMID8829945<br>Guinea pig anti Drosophila S6K (1:2000) was validated in the publication: PMID20444422<br>rabbit anti phospho-AMPKa(T172) (Cell signaling 50081) was validated in https://www.cellsignal.com/products/primary-antibodies/phospho-ampka-thr172-d4d6d-rabbit-mab/50081<br>rabbit anti-AMPKa (Cell signaling 2532S) 1:1000 was validated in https://www.cellsignal.com/products/primary-antibodies/ampka-antibody/2532<br>rabbit anti-atubulin (Cell signaling 2125) 1:1000 was validated in https://www.cellsignal.com/products/primary-antibodies/a-tubulin-11h10-rabbit-mab/2125 |

# Animals and other research organisms

Policy information about studies involving animals; ARRIVE guidelines recommended for reporting animal research, and Sex and Gender in Research

| | |
|---|---|
| Laboratory animals | Species: Drosophila melanogaster<br>Strains: Full genotype for each figure panel is provided in Suppl. Table 3. |
| Wild animals | No wild animals were used. |
| Reporting on sex | Sex was generally not considered in the study design |
| Field-collected samples | This study did not involve samples collected from the field. |
| Ethics oversight | This study does not require an ethical approval. |

Note that full information on the approval of the study protocol must also be provided in the manuscript.

# Plants

| | |
|---|---|
| Seed stocks | Not applicable. |
| Novel plant genotypes | Not applicable. |
| Authentication | Not applicable. |

