## [Peer Review File · Nature Cell Biology]

Remote-control of AMPK via extracellular adenosine controls tissue growth

Corresponding Author: Dr Aurelio Teleman

Version 0:

Decision Letter:

*Please delete the link to your author homepage if you wish to forward this email to co-authors.

Dear Dr Teleman,

Your manuscript, "Remote-control of AMPK via extracellular adenosine controls tissue growth", has now been seen by 3 referees, who are experts in *Drosophila* tissue metabolism (referee 1); AMPK signalling (referee 2); and tissue growth (referee 3). As you will see from their comments (attached below) they find this work of potential interest, but have raised substantial concerns, which in our view would need to be addressed with considerable revisions before we can consider publication in Nature Cell Biology.

Nature Cell Biology editors discuss the referee reports in detail within the editorial team, including the chief editor, to identify key referee points that should be addressed with priority, and requests that are overruled as being beyond the scope of the current study. To guide the scope of the revisions, I have listed these points below. We are committed to providing a fair and constructive peer-review process, so please feel free to contact me if you would like to discuss any of the referee comments further.

In particular, it would be essential to:

A- Perform additional experiments to strengthen the proposed model and clarify the interplay between AMPK, mTOR, and ecdysone signaling (reviewer#1's points (bullet pt 1-), Reviewer#1 pts 1-2, Reviewer#3 pt 1) and provide clarification on the discrepancy highlighted by reviewer#1 in the experiments on lowered yeast concentration.

B- Further test the mechanism by which dietary protein modulates Adgf-A levels (Reviewer#1 bullet pt 5, Reviewer#2 pt3, reviewer#3 pt 2))

C- All other referee concerns pertaining to strengthening existing data, methodological details, clarifications and textual changes, should also be addressed.

D- Finally please pay close attention to our guidelines on statistical and methodological reporting (listed below) as failure to do so may delay the reconsideration of the revised manuscript. In particular please provide:

We would be happy to consider a revised manuscript that would satisfactorily address these points, unless a similar paper is published elsewhere, or is accepted for publication in Nature Cell Biology in the meantime.

- ensure that it conforms to our format instructions and publication policies (see below and <https://www.nature.com/nature/for-authors>).

- provide a point-by-point rebuttal to the full referee reports verbatim, as provided at the end of this letter.

- provide the completed Reporting Summary (found here <https://www.nature.com/documents/nr-reporting-summary.pdf>). This is essential for reconsideration of the manuscript will be available to editors and referees in the event of peer review. For more information see <http://www.nature.com/authors/policies/availability.html> or contact me.

Nature Cell Biology is committed to improving transparency in authorship. As part of our efforts in this direction, we are now requesting that all authors identified as 'corresponding author' on published papers create and link their Open Researcher and Contributor Identifier (ORCID) with their account on the Manuscript Tracking System (MTS), prior to acceptance. ORCID helps the scientific community achieve unambiguous attribution of all scholarly contributions. You can create and link your ORCID from the home page of the MTS by clicking on 'Modify my Springer Nature account'. For more information please visit www.springernature.com/orcid.

This journal strongly supports public availability of data. Please place the data used in your paper into a public data repository, or alternatively, present the data as Supplementary Information. If data can only be shared on request, please explain why in your Data Availability Statement, and also in the correspondence with your editor. Please note that for some data types, deposition in a public repository is mandatory - more information on our data deposition policies and available repositories appears below.

Link Redacted

We would like to receive a revised submission within six months.

We hope that you will find our referees' comments, and editorial guidance helpful. Please do not hesitate to contact me if there is anything you would like to discuss.

Best wishes,

Sabrya Carim

Sabrya Carim, PhD
(she/her/hers)
Senior Editor, Nature Cell Biology
Nature Portfolio

Springer Nature
The Campus, 4 Crinan Street, London N1 9XW, UK
sabrya.carim@springernature.com
<https://orcid.org/0000-0001-9485-1938>

Reviewers' Comments:

Reviewer #1 (Remarks to the Author):

AMPK has traditionally been viewed as a stress-activated kinase. The authors here demonstrate that it can be activated by a signaling metabolite (adenosine) under non-stress conditions during development, revealing a novel regulatory mechanism

for AMPK activity. The finding that AMPK in one tissue (e.g., the wing disc) is regulated remotely by the intestine via circulating adenosine highlights an interesting interorgan communication pathway, relevant given the growing interest in systemic metabolic regulation and interorgan signaling. Furthermore, showing that dietary protein influences AMPK activity via a specific metabolic pathway (adenosine production) provides mechanistic insight into how nutrient availability modulates tissue growth. The observation that approximately half of dietary protein's effect on growth can be attributed to this pathway is a potentially very interesting finding of broad physiological relevance.

I have some points that should be addressed before publication to strengthen certain parts:

- Testing epistasis between certain components would strengthen the model that ecdysone signaling in the gut acts through Adgf-A to regulate circulating levels of adenosine, which then acts on the wing discs via AMPK to modulate TOR-dependent growth. Testing relationships between EcR and Adgf-A in enterocytes by knocking down or blocking EcR while overexpressing Adgf-A would help clarify whether gut ecdysone signaling acts through adenosine release (Adgf-A) in regulating systemic growth. Furthermore, while the data suggest that adenosine affects the TOR pathway, it is important to demonstrate that TOR mediates the growth effects of adenosine/AMPK in the wing disc. Genetic manipulations, such as partial knockdown or overactivation of TOR, in conjunction with altered AMPK activity, would provide stronger evidence that the gut signals through adenosine to AMPK, which then regulates growth via TOR. These experiments should also be extended to gut ecdysone signaling and Adgf-A to confirm that ecdysone production in the gut influences wing disc AMPK and TOR activity.
- Measuring larval weight at various developmental stages, rather than only pupariation timing and adult wing size, would clarify whether adenosine directly suppresses larval growth. This is particularly important as adult wing size often converges despite significant differences in larval growth rates and developmental timing.
- Examining whether adenosine also affects the growth of other larval-specific tissues, such as the fat body and prothoracic gland (PG), would provide further insight. If adenosine impacts TOR in the PG, it may reduce ecdysone production, potentially altering wing disc growth indirectly.
- The discrepancy between the experiments on lowered yeast concentration, where overexpressing Adgf-A does not accelerate development on high yeast, and the result in Figure 5c, where overexpression appears to accelerate development, needs clarification. Measuring larval growth rates under these conditions, rather than just pupariation timing, would support conclusions. Including Adgf-A loss-of-function analyses under varying yeast concentrations to determine whether the high-yeast growth effect is suppressed when Adgf-A is absent or reduced would strengthen these findings. A range of yeast concentrations with a two-way ANOVA to test interaction effects could also further support these conclusions. This is important as one of the main conclusions is that the mechanism accounts for half the dietary protein's influence on growth. In this regard, it is also important to note that while yeast is considered a main source of dietary protein, it also contains other nutrients, so perhaps down tone the language or show this with a pure protein source.
- The mechanism by which dietary yeast modulates Adgf-A levels in enterocytes remains unclear. Demonstrating whether yeast-derived signals, such as specific amino acids, directly regulate Adgf-A transcription or translation via, for example, TOR or GCN2 would help connect dietary protein intake to adenosine release.
- Confirming whether Adgf-A is exclusively expressed in enterocytes or is present in other gut cell types through tissue-specific expression data (e.g., reporters, IHC, or in situ hybridization) would clarify which cells contribute most to circulating adenosine production.
- Figure 3b should clearly indicate what is plotted on the y-axis.
- In Figure 4a, pS6 is normalized to Tubulin, whereas in 4b, pAMPK is normalized to pan-AMPK. Both pS6 and pAMPK should ideally be reported relative to their total protein (pan-S6 or pan-AMPK) as well as a loading control (e.g., Tubulin or Actin) for consistent and reliable quantification.
- Statistical tests should be clarified in all figure legends. The student t-test, which is currently used, can only be applied for pairwise comparisons on normally distributed data. In several cases, this should be changed to ANOVA followed by a multiple comparisons test. Furthermore, it seems that some of the data may not be normally distributed, so non-parametric tests, such as the Mann-Whitney U test for pairwise comparisons or the Kruskal-Wallis test for multiple comparisons, should be used. This must be corrected.
- Sample sizes (N) should be consistently indicated across all figures.
- Artwork such as wing discs could be improved and maybe also lighter colors would make it easier to see black text.

Overall, the discovery that adenosine produced in the intestine can remotely activate AMPK in wing discs under non-stress conditions is a key finding of broad interest. Additional experiments clarifying the interplay between AMPK, mTOR, and ecdysone signaling, as well as how dietary protein precisely regulates Adgf-A, will strengthen and further support the proposed model and highlight the physiological relevance of adenosine as a systemic growth regulator. I support the publication of this work after a revision.

Reviewer #2 (Remarks to the Author):

in this manuscript, Zhao et al develop a system to investigate the differential effects of the molting hormone ecdysone on growth regulation in the *Drosophila* wing disk. By combining *lexA*- and *GAL4* based manipulations, they can distinguish a cell autonomous and a non-autonomous effect. They confirm previous observations that autonomously, ecdysone can promote growth, and then go on to identify a second growth-inhibitory effect that is non-autonomous. This effect is mediated by gut enterocytes that secrete adenosine deaminase in an ecdysone-regulated manner to modulate hemolymph levels of adenosine, which had previously been identified as a growth inhibitor for wing disks. Zhao et al present data suggesting that adenosine does not act as a ligand for the adenosine receptor in the disk, but instead is taken up intracellularly and phosphorylated to yield AMP. AMP then can activate the AMP-dependent kinase, AMPK, which in turn inhibits TORC1 to stunt growth. In addition to ecdysone, adenosine deaminase expression in the gut is regulated by dietary cues, and changing expression levels there has effects on developmental timing at the whole animal level.

Overall, this is an impressive manuscript that combines several unexplained observations to come up with an interesting new mechanism of growth control. The method to dissect autonomous vs non-autonomous ecdysone effects is clever, and the technical quality of the experiments is overall high. as the manuscript aims to combine multiple distinct leads into one mechanism (i. e., identification of involved tissues, adenosine as a growth regulator, AMPK regulation by adenosine etc), some of the proposed links could be tied together better, and I have some technical comments.

1. The idea proposed is that adenosine phosphorylation increases AMP levels in the wing disc, but AMP levels are only shown for different timepoints for a control genotype in ED Fig. 4 A, known already from/consistent with Yamada et al., *Nat Metabolism* 2020 and Merkey et al., *J Insect Phys* 2011. This could also be an effect of metabolic changes in response to ecdysone. It would be nice to know whether *Adk2/3* are really the mediators of this - would mean that AMP can be produced also in the absence of metabolic changes (such as massive ATP use, or a decrease in ATP production), and would have potential implications for ATP/AMP buffering mechanisms.
2. the implication is that adenosine and AMPK inhibit proliferation via the TORC1 pathway, and indeed *TSC2* knockdown alleviates the ecdysone inhibitory effect. I am wondering if the latter effect could be unrelated. is there a way to show a direct link between AMPK and TOR in this system, e. g. through epistasis or Western blot?
3. There is still a gap in the argument regarding the adenosine uptake into the wing disc. this had been proposed to be mediated by *CNT/ENT* transporters in ref 40. Could a potential role be tested *in vitro*/pharmacologically?
4. statistical analyses are all t-test. This is certainly fine in most cases even though some samples do not quite seem to fit a normal distribution (Fig. 1B). Use of a paired t-test for Westerns in Fig 4 does not seem warranted, these are distinct unrelated samples. Can you comment?
5. Since all genetic manipulations rely on RNAi, it would be helpful to confirm some of the observed genotypes with an independent RNAi line. E. g., is the AMPK gamma subunit required that senses nucleotides?

Reviewer #3 (Remarks to the Author):

Zhang et al explore the interplay between ecdysone signalling and adenosine metabolism in regulating imaginal tissue growth in *Drosophila*. The authors provide compelling evidence that the steroid hormone ecdysone affects imaginal tissue proliferation both autonomously and non-autonomously through its effects on intestinal enterocytes. In particular, the authors show that ecdysone inhibits the production by intestinal enterocytes of *Adgf-A* a secreted enzyme that in turn depletes circulating adenosine pools. Adenosine depletion allows imaginal disc cells to proliferate by relieving the activation of the metabolic sensor AMPK, which is allosterically activated by adenosine conversion to AMP inside wing disc cells. Excitingly, this "remote-control" of imaginal disc cell proliferation by ecdysone via an intestinal relay is sensitive to dietary amino acids, providing a link between nutrient uptake and hormonal control of proliferation. The fact that ecdysone promotes imaginal disc cell proliferation tissue autonomously (shown here and in previous publications) and inhibits it with a delay via enterocytes and adenosine (this manuscript) opens the exciting possibility suggested by the authors that ecdysone pulses license successive pulses of growth throughout the developing animal that could be responsible for the coupling between growth and developmental progression. This studies therefore advances our understanding of developmental growth and opens up a rich area for future studies.

The data are generally compelling and well presented. There are a few mainly technical issues to be addressed in order to bolster the authors' conclusions as detailed below. With these points solidly explored, I believe this manuscript would have substantial impact in the field and potentially wide implications for our understanding of metabolic scaling, as well as tumour biology or other developmental growth disorders.

1) Specificity of *GAL4* Drivers

The authors' tissue-specific knockdown experiments explore the non-autonomous effects of ecdysone signalling. However, as is usually the case some of the *GAL4* driver used are not completely specific in their expression patterns. It is important to strengthen the data implicating enterocytes as the source of *Adgf-A*, and exclude the ring gland and hemocytes in this context, particularly since prior work suggests that ring gland/hemocyte-expressed *Adgf-A* is sufficient to rescue developmental defects in *Adgf-A* mutants (Dolezal et al., 2005).

We therefore suggest the following experiments:

- Use the alternative enterocyte driver *mex1-GAL4* (BL91368) to repeat the experiments in Fig. 1f and 4a.
- Add the general ring gland driver *P0206-GAL4* and a hemocyte driver (*hml-GAL4* or *eater-GAL4*) to the panel in ED Fig. 2.

Ideally the authors could also check that *MyoIA* and *mex1-GAL4* are not significantly active in the L3 ring gland.

2) Mechanism of intestinal ecdysone signalling

Many possible mechanisms could explain how *Adgf-A* expression is activated by dietary amino acids in enterocytes, but the simplest to test would be mTORC1-dependent. The authors could increase/decrease mTORC1 activity in ECs (*Rheb/Tsc1/2* overexpression) and test the effect on *Adgf-A* expression.

METHODS – Nature Cell Biology publishes methods online. The methods section should be provided as a separate Word document, which will be copyedited and appended to the manuscript PDF, and incorporated within the HTML format of the

paper.

Methods should be written concisely, but should contain all elements necessary to allow interpretation and replication of the results. As a guideline, Methods sections typically do not exceed 3,000 words. The Methods should be divided into subsections listing reagents and techniques. When citing previous methods, accurate references should be provided and any alterations should be noted. Information must be provided about: antibody dilutions, company names, catalogue numbers and clone numbers for monoclonal antibodies; sequences of RNAi and cDNA probes/primers or company names and catalogue numbers if reagents are commercial; cell line names, sources and information on cell line identity and authentication. Animal studies and experiments involving human subjects must be reported in detail, identifying the committees approving the protocols. For studies involving human subjects/samples, a statement must be included confirming that informed consent was obtained. Statistical analyses and information on the reproducibility of experimental results should be provided in a section titled "Statistics and Reproducibility".

All Nature Cell Biology manuscripts submitted on or after March 21 2016 must include a Data availability statement as a separate section after Methods but before references, under the heading "Data Availability". For Springer Nature policies on data availability see <http://www.nature.com/authors/policies/availability.html>; for more information on this particular policy see <http://www.nature.com/authors/policies/data/data-availability-statements-data-citations.pdf>. The Data availability statement should include:

- Accession codes for primary datasets (generated during the study under consideration and designated as "primary accessions") and secondary datasets (published datasets reanalysed during the study under consideration, designated as "referenced accessions"). For primary accessions data should be made public to coincide with publication of the manuscript. A list of data types for which submission to community-endorsed public repositories is mandated (including sequence, structure, microarray, deep sequencing data) can be found here <http://www.nature.com/authors/policies/availability.html#data>.
- Unique identifiers (accession codes, DOIs or other unique persistent identifier) and hyperlinks for datasets deposited in an approved repository, but for which data deposition is not mandated (see here for details <http://www.nature.com/sdata/data-policies/repositories>).
- At a minimum, please include a statement confirming that all relevant data are available from the authors, and/or are included with the manuscript (e.g. as source data or supplementary information), listing which data are included (e.g. by figure panels and data types) and mentioning any restrictions on availability.
- If a dataset has a Digital Object Identifier (DOI) as its unique identifier, we strongly encourage including this in the Reference list and citing the dataset in the Methods.

We recommend that you upload the step-by-step protocols used in this manuscript to protocols.io. More details can be found at <https://www.protocols.io/help/publish-articles>.

All imaging data should be accompanied by scale bars, which should be defined in the legend.

Cropped images of gels/blots are acceptable, but need to be accompanied by size markers, and to retain visible background signal within the linear range (i.e. should not be saturated). The boundaries of panels with low background have to be demarked with black lines. Splicing of panels should only be considered if unavoidable, and must be clearly marked on the figure, and noted in the legend with a statement on whether the samples were obtained and processed simultaneously. Quantitative comparisons between samples on different gels/blots are discouraged; if this is unavoidable, it should only be performed for samples derived from the same experiment with gels/blots were processed in parallel, which needs to be stated in the legend.

- For line art, graphs, charts and schematics we prefer Adobe Illustrator (.AI), Encapsulated PostScript (.EPS) or Portable Document Format (.PDF). Files should be saved or exported as such directly from the application in which they were made,

to allow us to restyle them according to our journal house style.

The total number of Supplementary Figures (not including the "unprocessed scans" Supplementary Figure) should not exceed the number of main display items (figures and/or tables (see our Guide to Authors and March 2012 editorial <http://www.nature.com/ncb/authors/submit/index.html#suppinfo>; <http://www.nature.com/ncb/journal/v14/n3/index.html#ed>). No restrictions apply to Supplementary Tables or Videos, but we advise authors to be selective in including supplemental data.

GUIDELINES FOR EXPERIMENTAL AND STATISTICAL REPORTING

REPORTING REQUIREMENTS – We are trying to improve the quality of methods and statistics reporting in our papers. To that end, we are now asking authors to complete a reporting summary that collects information on experimental design and reagents. The Reporting Summary can be found here <https://www.nature.com/documents/nr-reporting->

summary.pdf"><https://www.nature.com/documents/nr-reporting-summary.pdf>)If you would like to reference the guidance text as you complete the template, please access these flattened versions at http://www.nature.com/authors/policies/availability.html.
STATISTICS – Wherever statistics have been derived the legend needs to provide the n number (i.e. the sample size used to derive statistics) as a precise value (not a range), and define what this value represents. Error bars need to be defined in the legends (e.g. SD, SEM) together with a measure of centre (e.g. mean, median). Box plots need to be defined in terms of minima, maxima, centre, and percentiles. Ranges are more appropriate than standard errors for small data sets. Wherever statistical significance has been derived, precise p values need to be provided and the statistical test used needs to be stated in the legend. Statistics such as error bars must not be derived from n<3. For sample sizes of n<5 please plot the individual data points rather than providing bar graphs. Deriving statistics from technical replicate samples, rather than biological replicates is strongly discouraged. Wherever statistical significance has been derived, precise p values need to be provided and the statistical test stated in the legend.
Information on how many times each experiment was repeated independently with similar results needs to be provided in the legends and/or Methods for all experiments, and in particular wherever representative experiments are shown.
We strongly recommend the presentation of source data for graphical and statistical analyses as a separate Supplementary Table, and request that source data for all independent repeats are provided when representative experiments of multiple independent repeats, or averages of two independent experiments are presented. This supplementary table should be in Excel format, with data for different figures provided as different sheets within a single Excel file. It should be labelled and numbered as one of the supplementary tables, titled "Statistics Source Data", and mentioned in all relevant figure legends.
----- Please don't hesitate to contact NCB@nature.com should you have queries about any of the above requirements ----
Version 1:
Decision Letter:
Our ref: NCB-A56464A
12th June 2025
Dear Dr. Teleman,
Thank you for submitting your revised manuscript "Remote-control of AMPK via extracellular adenosine controls tissue growth" (NCB-A56464A). It has now been seen by the original referees and their comments are below. The reviewers find that the paper has improved in revision, and therefore we'll be happy in principle to publish it in Nature Cell Biology, pending minor revisions to satisfy the referees' final requests and to comply with our editorial and formatting guidelines.
Please ensure to address the reviewers' final points in the revised manuscript. Please ensure that all figures fit into a single standard page and adhere to a maximum page size of roughly 180mm wide x 200mm high. To ensure legibility once figures are re-sized, please use a font size of no smaller than 6pt Arial or Helvetica throughout the figures.
If the current version of your manuscript is in a PDF format, please email us a copy of the file in an editable format (Microsoft Word or LaTeX)-- we can not proceed with PDFs at this stage.
We are now performing detailed checks on your paper and will send you a checklist detailing our editorial and formatting requirements in about two weeks.
**Please do not upload the final materials and make any revisions until you receive this additional information from us.**
Thank you again for your interest in Nature Cell Biology Please do not hesitate to contact me if you have any questions.
Sincerely,
Sabrya Carim, PhD
(she/her/hers)
Senior Editor, Nature Cell Biology
Nature Portfolio
Springer Nature
The Campus, 4 Crinan Street, London N1 9XW, UK
sabrya.carim@springernature.com
<https://orcid.org/0000-0001-9485-1938>
</html>

Reviewer #1 (Remarks to the Author):

The authors have provided a thorough and careful revision of the manuscript, addressing the concerns raised. The additional experiments substantially strengthen the conclusions and provide strong support for the proposed model. The inclusion of GCN2/ATF4-dependent regulation of Adgf-A expression further clarifies how dietary amino acids control this pathway. The statistical analyses have also been carefully described. That said, I note that several of the figures still lack statistical annotations, such as those showing growth rates and pupariation timing (ED Fig. 6), ED Fig. 7e where ecdysone-inducible gene expression is measured, and Reviewer Fig. 9 presenting Adgf-A mRNA levels under different dietary conditions. Nevertheless, these points do not affect the overall conclusions of the manuscript but should be addressed. In summary, the authors have addressed the previous concerns in a satisfactory manner. The new data convincingly support the central claims, and the revised manuscript is suitable for publication after minor attention to the remaining points regarding missing statistical annotations.

Reviewer #2 (Remarks to the Author):

In this manuscript, Zhao et al develop a system to distinguish a cell autonomous and a non-autonomous effect of ecdysone on wing disk growth, identifying a second growth-inhibitory effect that is non-autonomous. This effect is mediated by gut enterocytes that secrete Adgf to modulate hemolymph levels of adenosine, which had previously been identified as a growth inhibitor for wing disks. They suggest that adenosine is taken up intracellularly and phosphorylated to yield AMP. AMP then can activate the AMP-dependent kinase, AMPK, which in turn inhibits TORC1 to stunt growth. In addition to ecdysone, adenosine deaminase expression in the gut is regulated by dietary cues, and changing expression levels there has effects on developmental timing at the whole animal level.

In my original review, I asked to assess whether Adks can induce real AMP concentration changes in the wing disc, to provide further evidence that AMPK is upstream of TOR in the Adgf pathway, to assess potential roles for nucleoside transporters in adenosine uptake, to reassess their use of statistic tests and to confirm key experiments with independent RNAi lines. The authors have addressed these questions convincingly, and the experiments support the proposed model. I am particularly impressed that one can see the effect of Adk knockdown on AMP concentrations.

Reviewer #3 (Remarks to the Author):

The authors have addressed my comments in the revised version and I am happy to support publication in Nature Cell Biology.

Version 2:

Decision Letter:

Dear Dr Teleman,

I am pleased to inform you that your manuscript, "Remote-control of AMPK via extracellular adenosine controls tissue growth", has now been accepted for publication in Nature Cell Biology. Congratulations!

Please note that *Nature Cell Biology* is a Transformative Journal (TJ). Authors may publish their research with us through the traditional subscription access route or make their paper immediately open access through payment of an article-processing charge (APC). Authors will not be required to make a final decision about access to their article until it has been accepted. [Find out more about Transformative Journals](https://www.springernature.com/gp/open-research/transformative-journals)

Authors may need to take specific actions to achieve compliance with funder and institutional open access mandates. If your research is supported by a funder that requires immediate open access (e.g. according to [Plan S principles](https://www.springernature.com/gp/open-science/plan-s-compliance) or the [NIH public access policy](https://www.springernature.com/gp/open-science/us-federal-agency-compliance)) then you should select the gold OA route, and we will direct you to the compliant route where possible. Because authors warrant under our subscription licensing terms that they haven't committed to licensing any version of their article under a licence inconsistent with the terms of our agreement – including the applicable embargo period – publication under the subscription model isn't suitable for authors whose funders require no embargo.

If you have not already done so, we strongly recommend that you upload the step-by-step protocols used in this manuscript to protocols.io (<https://protocols.io>), an open online resource that allows researchers to share their detailed experimental know-how. All uploaded protocols are made freely available and are assigned DOIs for ease of citation. Protocols and Nature Portfolio journal papers in which they are used can be linked to one another, and this link is clearly and prominently visible in the online versions of both. Authors who performed the specific experiments can act as primary authors for the Protocol as they will be best placed to share the methodology details, but the Corresponding Author of the present research paper should be included as one of the authors. By uploading your Protocols onto protocols.io, you are enabling researchers to more readily reproduce or adapt the methodology you use, as well as increasing the visibility of your protocols and papers. You can also establish a dedicated workspace to collect your lab Protocols. Further information can be found at <https://www.protocols.io/help/publish-articles>.

Nature Cell Biology encourages authors presenting evidence for cell, biological, molecular, and genetic interactions to consider communicating these findings using Biofactoid (<https://biofactoid.org/>). This tool helps users share a searchable representation of interactions (e.g. binding, gene expression, post-translational modification) between genes, gene products, or chemicals. Information added to Biofactoid, with author attribution, is shared on social media and public databases, such as Pathway Commons, where it can be discovered and analyzed in the context of a large and growing corpus of knowledge.

With kind regards,

Sabrya Carim, PhD
(she/her/hers)
Senior Editor, Nature Cell Biology
Nature Portfolio

Springer Nature
The Campus, 4 Crinan Street, London N1 9XW, UK
sabrya.carim@springernature.com
<https://orcid.org/0000-0001-9485-1938>

** Visit the Springer Nature Editorial and Publishing website at http://editorial-jobs.springernature.com?utm_source=ejp_NCB_email&utm_medium=ejp_NCB_email&utm_campaign=ejp_NCB for more information about our career opportunities. If you have any questions please click [here](mailto:editorial.publishing.jobs@springernature.com).

Reviewer #1

AMPK has traditionally been viewed as a stress-activated kinase. The authors here demonstrate that it can be activated by a signaling metabolite (adenosine) under non-stress conditions during development, revealing a novel regulatory mechanism for AMPK activity. The finding that AMPK in one tissue (e.g., the wing disc) is regulated remotely by the intestine via circulating adenosine highlights an interesting interorgan communication pathway, relevant given the growing interest in systemic metabolic regulation and interorgan signaling. Furthermore, showing that dietary protein influences AMPK activity via a specific metabolic pathway (adenosine production) provides mechanistic insight into how nutrient availability modulates tissue growth. The observation that approximately half of dietary protein's effect on growth can be attributed to this pathway is a potentially very interesting finding of broad physiological relevance.

We thank the reviewer for the positive evaluation and the constructive suggestions below. We hope to have addressed all the issues raised by the reviewer.

I have some points that should be addressed before publication to strengthen certain parts:

- Testing epistasis between certain components would strengthen the model that ecdysone signaling in the gut acts through Adgf-A to regulate circulating levels of adenosine, which then acts on the wing discs via AMPK to modulate TOR-dependent growth. Testing relationships between EcR and Adgf-A in enterocytes by knocking down or blocking EcR while overexpressing Adgf-A would help clarify whether gut ecdysone signaling acts through adenosine release (Adgf-A) in regulating systemic growth.

We thank the reviewer for the suggestion. We did 2 experiments to address this issue.

The first experiment, done in the spok-i system, is challenging because it requires many simultaneous manipulations: 1) removing ecdysone signaling globally, so that we can control its activation in enterocytes, 2) activating ecdysone signaling in the wing disc, so it is capable of proliferating, and then for the epistasis 3) +/- turning on ecdysone signaling in enterocytes, and 4) +/- turning on Adgf-A expression in enterocytes. To achieve this, we used the LexA/LexAop system to knock down spok, and we used C765-GAL4 to knock down EcR in wing discs to autonomously induce wing disc proliferation (Fig.1b). We then knocked down EcR also in enterocytes by including Myo1a-GAL4, which causes wing disc proliferation to terminate by 96h (Fig.1f). We then added-in a UAS-Adgf-A transgene, and as expected, this reversed the proliferation termination, enabling wing discs to once again proliferate at 96h. (Reviewer Fig.1 and ED Fig. 3e). This is essentially the result the reviewer was asking

for. The reason this is not a perfect experiment is that Adgf-A is not only expressed in enterocytes. However, please note that anyways Adgf-A is a secreted enzyme that circulates in the hemolymph, so it does not really matter which tissue makes it. Furthermore, the complementary experiment to address this same epistasis which we had in the original manuscript used ecdysone feeding to activate ecdysone in both the wing and the enterocytes, and in this case this "freed" the GAL4 system so that we could use it to specifically express Adgf-A in enterocytes, and this also enabled wing disc proliferation at 96h (Figure 2d).

In a second experiment, we measured pupation timing of animals upon knock-down of EcR and overexpression of Adgf-A only in enterocytes with Myo1a-GAL4 in an otherwise wildtype background. As expected, knockdown of EcR in enterocytes (which reduces Adgf-A expression) causes a pupation delay (Reviewer Figure 2, ED Fig. 6b), and this delay is rescued by Adgf-A expression (Reviewer Figure 2, ED Fig. 6b).

Together, these results show that the inhibitory effect of ecdysone signaling in enterocytes on growth is mediated by Adgf-A.

Furthermore, while the data suggest that adenosine affects the TOR pathway, it is important to demonstrate that TOR mediates the growth effects of adenosine/AMPK in the wing disc. Genetic manipulations, such as partial knockdown or overactivation of TOR, in conjunction with altered AMPK activity, would provide stronger evidence that the gut signals through adenosine to AMPK, which then regulates growth via TOR.

As suggested by the reviewer, we did an experiment where we partially knocked down TOR activity in conjunction with altered AMPK activity, to show that the proliferative effect is mediated by TOR. As background to set up the experiment: Figure 2d shows that feeding ecdysone to *spok-i* larvae causes wing discs to stop proliferating at 96 hours. This is due to elevated adenosine, since it is rescued by *Adgf-A* expression. The effect of this elevated adenosine is mediated by AMPK (because AMPK knockdown in the wing rescues this effect, Figure 4d) and by TOR (because it is rescued by TOR activation via *TSC2* knockdown, Figure 4e). As suggested, we now added an epistatic experiment which shows that TOR mediates the effect of AMPK on proliferation (Reviewer Figure 3, ED Fig. 5d): although knockdown of AMPK enables the wing discs to proliferate at 96h, simultaneous knockdown of either *Rheb* or *Raptor* reverses this effect, showing that the effect of AMPK on proliferation is mediated by TOR.

These experiments should also be extended to gut ecdysone signaling and *Adgf-A* to confirm that ecdysone production in the gut influences wing disc AMPK and TOR activity.

To address this, we knocked-down *EcR* in enterocytes with *Myo1A-GAL4* and then immunoblotted wing disc lysates to assay AMPK and TOR activity (Reviewer Figure

4, ED Fig. 4f). As expected, this shows that EcR-RNAi in enterocytes leads to elevated AMPK activity and reduced TOR activity in the wing disc. (Note that we also included total S6 in this blot, as requested by the reviewer below.)

Regarding the effect of gut Adgf-A on wing disc AMPK and TOR activity, please note this was already in the manuscript: Figures 4a and 4b show that Adgf overexpression or knockdown in enterocytes regulates AMPK and TOR activity in wing discs, in both spok-i and wildtype backgrounds, respectively. We now additionally show that the effect of circulating adenosine on wing TOR activity is mediated by AMPK. New ED Figure 7c (Reviewer Figure 5) shows that feeding adenosine causes a reduction in wing TOR activity, and this reduction is gone upon wing-disc specific knockdown of AMPK. (Here too we included total S6 in the blot).

- Measuring larval weight at various developmental stages, rather than only pupariation timing and adult wing size, would clarify whether adenosine directly suppresses larval growth.

This is particularly important as adult wing size often converges despite significant differences in larval growth rates and developmental timing.

We now measured L3 larval weight until pupation and find that indeed adenosine not only regulates imaginal disc growth but also larval growth. Modulation of circulating adenosine by either overexpressing or knocking down Adgf-A in enterocytes causes larvae to grow larger and faster, or smaller and slower, respectively (Reviewer Fig.6, ED Fig. 6a)

- Examining whether adenosine also affects the growth of other larval-specific tissues, such as the fat body and prothoracic gland (PG), would provide further insight. If adenosine impacts TOR in the PG, it may reduce ecdysone production, potentially altering wing disc growth indirectly.

From the data presented in Reviewer Figure 6 above, it does indeed appear that adenosine regulates growth of larval tissues as well as imaginal tissues, since larval weight is strongly affected (almost 2-fold).

To test whether adenosine impacts ecdysone production by the PG, we fed larvae with exogenous adenosine (which affects AMPK and TOR activity in wing discs, ED Fig. 7b, as well as pupation rate, ED Fig. 7d) for 24 hours and then analyzed expression by RT-PCR of ecdysone target genes Eip71CD, Eip74B and ftz-f1, which sensitively read out ecdysone signaling (Reviewer Fig. 7, ED Fig. 7e). Note that Eip71CD and Eip74B are induced by ecdysone signaling whereas ftz-f1 is inhibited. We found no indication that adenosine reduces ecdysone signaling. If anything, Eip71CD expression increases, but this does not happen to Eip74B, nor is there a drop in ftz-f1 expression, suggesting that increased expression of Eip71CD is not caused by ecdysone. In sum, we find no evidence for adenosine reducing ecdysone production by the PG.

(In addition, please note that in the spok-i system, endogenous ecdysone synthesis is removed, so even if adenosine were capable of modulating ecdysone synthesis, this could not account for the effects we observe.)

- The discrepancy between the experiments on lowered yeast concentration, where overexpressing Adgf-A does not accelerate development on high yeast, and the result in Figure 5c, where overexpression appears to accelerate development, needs clarification.

This is a good point. The reason is that the "high-yeast" food, where Adgf-A expression does not accelerate development, is more rich than our standard food, where Adgf-A expression does accelerate development. We have now included as Fig. ED 6h (Reviewer Figure 8 below) a pupation curve to show this is the case, and have added an explanation to the Results section.

Measuring larval growth rates under these conditions, rather than just pupariation timing, would support conclusions.

We have now added larval growth rates to the pupation curves (ED Fig. 6a, ED Fig. 6i), which show that adenosine not only inhibits the rate of imaginal tissue growth, but also inhibits the rate of larval growth and the final size of the larva at pupation.

Including Adgf-A loss-of-function analyses under varying yeast concentrations to determine whether the high-yeast growth effect is suppressed when Adgf-A is absent or reduced would strengthen these findings.

We have tried this experiment but unfortunately it is not interpretable for technical reasons. Although the Adgf-A knockdown is very efficient (compare blue and green bars in the "gut" samples in Reviewer Figure 9), the high yeast is nonetheless still able to induce its expression (pink vs green bars). Therefore the genetic epistasis cannot be done in this direction.

A range of yeast concentrations with a two-way ANOVA to test interaction effects could also further support these conclusions. This is important as one of the main conclusions is that the mechanism accounts for half the dietary protein's influence on growth.

We have now done pupation curves (and larval weight measurements) on two genotypes (control vs Adgf-A overexpression) versus three dietary yeast concentrations (+0%, +5% and +10% yeast), then extracted the timepoints of 50% pupation, and used these values to perform a 2-way ANOVA (ED Fig. 6i, Reviewer Figure 10). The analysis confirmed that the effect of yeast is partly dependent on Adgf-A with a high statistical significance (i.e. p value for the interaction term <0.0001).

Reviewer Figure 10: Dietary yeast affects growth rate in part via regulation of adenosine levels. Pupation curves of control or Adgf-A overexpressing animals on food supplemented with three different yeast concentrations. First instar larvae of indicated genotypes were seeded at equal density in food supplemented with 10%, 5% or 0% yeast at 29°C to enable GAL4 expression, and then larval body weight (bottom) and pupation rate (top) were assayed twice per day. The timepoints (in days) of 50% pupation, which were used for the 2-way ANOVA, are indicated in the table on the bottom (derived from non-linear regression).

In this regard, it is also important to note that while yeast is considered a main source of dietary protein, it also contains other nutrients, so perhaps down tone the language or show this with a pure protein source.

We now provide data showing that dietary amino acids regulate Adgf-A expression (ED Fig. 6c) and that these are not sensed by the mTORC1 pathway (ED Fig. 6f) but by the GCN2-ATF4 pathway (ED Fig. 6g) (see the next point below with Reviewer Figure 11).

- The mechanism by which dietary yeast modulates Adgf-A levels in enterocytes remains unclear. Demonstrating whether yeast-derived signals, such as specific amino acids, directly regulate Adgf-A transcription or translation via, for example, TOR or GCN2 would help connect dietary protein intake to adenosine release.

We thank the reviewer for this suggestion. We now provide data showing that dietary amino acids regulate Adgf-A transcription via GCN2 and ATF4 (Reviewer Fig. 11). We assayed Adgf-A mRNA levels by Q-RT-PCR in intestines of control or knockdown larvae fed with food supplemented with or without essential amino acids (EAA). We find that constitutively activating mTORC1 by knocking down TSC2, or constitutively inactivating mTORC1 by knocking down Rheb or Raptor in enterocytes does not blunt the induction of Adgf-A in response to amino acids (Reviewer Fig. 11 top panel). Instead, knockdown of either GCN2 or the downstream transcription factor ATF4 strongly blunts this induction (lower panel Reviewer Figure 11). We think this is interesting because it provides a physiological function for GCN2, which senses the absence of any amino acid. These results add significant new mechanistic insight to the story and we thank the reviewer for having suggested this experiment.

Reviewer Figure 11: Dietary amino acids regulate Adgf-A via the GCN2-ATF4 pathway.

Adgf-A expression was quantified by Q-RT-PCR on RNA extracted from intestines of animals with the indicated genotypes.

Top: The induction of gut Adgf-A expression in response to dietary amino acids is not blunted if mTORC1 activity in intestinal enterocytes is either maximally induced by knockdown of TSC2 or maximally repressed by knockdown of either Raptor or Rheb.

Bottom: The induction of gut Adgf-A expression in response to dietary amino acids is blunted if either GCN2 or the downstream transcription factor ATF4 is knocked down in enterocytes.

- Confirming whether Adgf-A is exclusively expressed in enterocytes or is present in other gut cell types through tissue-specific expression data (e.g., reporters, IHC, or in situ hybridization) would clarify which cells contribute most to circulating adenosine production.

To determine Adgf-A expression within the gut, we used an endogenous knock-in Adgf-A-GFP reporter line, in which the Adgf-A coding sequence is replaced by GFP. Using this line, we found that Adgf-A (GFP) is expressed in the large polyploid enterocytes of the larval gut, but not in other cells. (in Reviewer Fig. 12).

Reviewer Figure 12: GFP from an Adgf-A-GFP reporter line, in which GFP replaces the endogenous Adgf-A coding sequence, is expressed in enterocytes.

This is consistent with data from our original manuscript, where we performed cell-type-specific knockdown of EcR using different GAL4 drivers that specifically target ISC, ECs, EEs or EBs in the gut (Fig. 1f) which showed that only EcR knockdown in enterocytes resulted in wing disc proliferation termination, while knockdown in stem cells and enteroendocrine cells it had no effect.

- Figure 3b should clearly indicate what is plotted on the y-axis.

Thanks - fixed !

- In Figure 4a, pS6 is normalized to Tubulin, whereas in 4b, pAMPK is normalized to pan-AMPK. Both pS6 and pAMPK should ideally be reported relative to their total protein (pan-S6 or pan-AMPK) as well as a loading control (e.g., Tubulin or Actin) for consistent and reliable quantification.

We have repeated Figure 4a using a second intestinal enterocyte driver, Mex1-GAL4, as requested by Reviewer 3 (ED Fig. 4e) and in this replicate we included total S6, confirming the result obtained in Fig. 4a. We also included total S6 in the other new immunoblots, such as in ED Fig. 4f (Reviewer Figure 4 above).

- Statistical tests should be clarified in all figure legends. The student t-test, which is currently used, can only be applied for pairwise comparisons on normally distributed data. In several cases, this should be changed to ANOVA followed by a multiple comparisons test. Furthermore, it seems that some of the data may not be normally distributed, so non-parametric tests, such as the Mann-Whitney U test for pairwise comparisons or the Kruskal-Wallis test for multiple comparisons, should be used. This must be corrected.

We carefully re-examined all statistical analyses. For each dataset, we first used the Shapiro-Wilk test to check for data normality. For data that followed a normal distribution, we used a two-tailed t-test for pairwise comparisons, and ANOVA with Dunnett's multiple comparison correction for multi-group comparisons. For data that did not follow normality, we used the two-tailed Mann-Whitney U test for pairwise comparisons, and the Kruskal-Wallis test with Dunn's post hoc correction for multi-group comparisons. The p-values are corrected and labelled in each figure, and the test that we used is indicated in the figure legend. This did not change any of the conclusion from the data.

- Sample sizes (N) should be consistently indicated across all figures.

Done. The sample sizes (n) are either written directly in the figure, or in the legend.

- Artwork such as wing discs could be improved and maybe also lighter colors would make it easier to see black text.

We have increased the lightness and reduced the saturation of the coloring to make the text more legible.

Overall, the discovery that adenosine produced in the intestine can remotely activate AMPK in wing discs under non-stress conditions is a key finding of broad interest. Additional experiments clarifying the interplay between AMPK, mTOR, and ecdysone signaling, as well as how dietary protein precisely regulates Adgf-A, will strengthen and further support the proposed model and highlight the physiological relevance of adenosine as a systemic growth regulator. I support the publication of this work after a revision.

We thank the reviewer for the supportive words, and all the experimental suggestions, which we think have improved the quality of the study.

Reviewer #2

in this manuscript, Zhao et al develop a system to investigate the differential effects of the molting hormone ecdysone on growth regulation in the drosophila wing disk. By combining lexA- and GAL4 based manipulations, they can distinguish a cell autonomous and a non-autonomous effect. They confirm previous observations that autonomously, ecdysone can promote growth, and then go on to identify a second growth-inhibitory effect that is non-autonomous. This effect is mediated by gut enterocytes that secrete adenosine deaminase in an ecdysone-regulated manner to modulate hemolymph levels of adenosine, which had previously been identified as a growth inhibitor for wing disks. Zhao et al present data suggesting that adenosine does not act as a ligand for the adenosine receptor in the disk, but instead is taken up intracellularly and phosphorylated to yield AMP. AMP then can activate the AMP-dependent kinase, AMPK, which in turn inhibits TORC1 to stunt growth. In addition to ecdysone, adenosine deaminase expression in the gut is regulated by dietary cues, and changing expression levels there has effects on developmental timing at the whole animal level.

Overall, this is an impressive manuscript that combines several unexplained observations to come up with an interesting new mechanism of growth control. The method to dissect autonomous vs non-autonomous ecdysone effects is clever, and the technical quality of the experiments is overall high. as the manuscript aims to combine multiple distinct leads into one mechanism (i. e., identification of involved tissues, adenosine as a growth

regulator, AMPK regulation by adenosine etc), some of the proposed links could be tied together better, and I have some technical comments.

We thank the Reviewer for the positive evaluation and for the constructive suggestions - we have performed the experiments and we believe this has strengthened the study.

1. The idea proposed is that adenosine phosphorylation increases AMP levels in the wing disc, but AMP levels are only shown for different timepoints for a control genotype in ED Fig. 4 A, known already from/consistent with Yamada et al., Nat Metabolism 2020 and Merkey et al., J Insect Phys 2011. This could also be an effect of metabolic changes in response to ecdysone. It would be nice to know whether Adk2/3 are really the mediators of this - would mean that AMP can be produced also in the absence of metabolic changes (such as massive ATP use, or a decrease in ATP production), and would have potential implications for ATP/AMP buffering mechanisms.

We thank the reviewer for this important point. As suggested, we knocked down both Adk2 and Adk3 specifically in the wing disc and measured AMP levels in wing discs from *spok* larvae fed with 20E for 96h, when AMP levels increase (ED Fig. 4a). This revealed a strong reduction in AMP levels in the Adk knockdown wing discs compared to controls (ED Fig. 4b and Reviewer Figure 13 below). AMP levels increase roughly 2.5-fold at 96h compared to 48h (ED Fig. 4a), and Adk knockdown causes them to drop roughly 2-fold. This indicates that AMP is indeed produced via adenosine phosphorylation, supporting a mode of AMP production that is not due to ATP metabolic changes.

2. the implication is that adenosine and AMPK inhibit proliferation via the TORC1 pathway, and indeed TSC2 knockdown alleviates the ecdysone inhibitory effect. I am wondering if the latter effect could be unrelated. is there a way to show a direct link between AMPK and TOR in this system, e. g. through epistasis or Western blot?

As suggested, to test if adenosine inhibits mTORC1 via AMPK, we fed adenosine to larvae and then performed an immunoblot on lysates from wing discs (first two lanes, Reviewer Figure 14 and ED Fig. 7c). This revealed that, as expected, adenosine activates AMPK and inhibits mTORC1. Importantly, we also immunoblotted lysates from wing discs where AMPK was specifically knocked down in the wing discs (3rd and 4th lanes) and found that this alleviates the effect of adenosine on mTORC1. Thus the inhibitory effect of adenosine on mTORC1 is via AMPK.

3. There is still a gap in the argument regarding the adenosine uptake into the wing disc. this had been proposed to be mediated by CNT/ENT transporters in ref 40. Could a potential role be tested in vitro/pharmacologically?

Since there are multiple CNT and ENT transporters, we tested this by inhibiting the transporters pharmacologically in wing disc explants (Reviewer Figure 15, ED Fig. 5a). First we confirmed that removal of adenosine deaminase from the explant medium causes wing discs to stop proliferating, indicating that adenosine is blocking proliferation. We then added the ENT inhibitor dipyridamole (30µM), and found that this enables wing discs to proliferate also in the absence of adenosine deaminase, indicating that indeed the adenosine needs to enter into the cells via the CNT/ENT transporters to block proliferation. Note that addition of dipyridamole to explants in the +ADA condition (where adenosine is removed) causes proliferation to decrease somewhat (compare bars 3 and 4 in the EdU quantification, Reviewer Figure 15). This is likely because the CNT/ENT transporters are not specific for adenosine, but transport nucleosides more generally, and therefore blocking them makes the cells a bit unhappy. This probably explains why addition of dipyridamole does not bring disc proliferation levels up to the same levels as adding adenosine deaminase (compare bars 2 and 3), but rather to the levels of adenosine deaminase + dipyridamole (lanes 2 and 4).

Reviewer Figure 15: Extracellular adenosine inhibits wing disc proliferation via ENT/CNT transporters. Wing discs cultured ex vivo in medium containing calf intestinal adenosine deaminase (ADA), ENT/CNT inhibitor dipyridamole, or neither, for 3 hours and then stained with EdU. (Top) representative images. (Bottom) quantification of EdU.

4. statistical analyses are all t-test. This is certainly fine in most cases even though some samples do not quite seem to fit a normal distribution (Fig. 1B). Use of a paired t-test for Westerns in Fig 4 does not seem warranted, these are distinct unrelated samples. Can you comment?

In response also to a comment by Reviewer 1, we carefully re-examined all statistical analyses. For each dataset, we first used the Shapiro-Wilk test to check for data normality. For data that followed a normal distribution, we used a two-tailed t-test for pairwise comparisons, and ANOVA with Dunnett's multiple comparison correction for multi-group comparisons. For data that did not follow normality, we used the two-tailed Mann-Whitney U test for pairwise comparisons, and the Kruskal-Wallis test with Dunn's post hoc correction for multi-group comparisons. The p-values are corrected and labelled in each figure, and the test that we used is indicated in the figure legend. This did not change any of the conclusion from the data.

For the westerns in Figure 4, we use a paired test because the experiment was done on multiple different days, and each day had slightly different experimental conditions, blot exposure time, etc. hence there are small uncontrolled parameters which simultaneously influence the intensities of all the genotypes/bands in one replicate. In such cases a paired test can be used to account for such factors that influence all the samples in a pairing. (This is analogous to doing a before/after drug treatment measurement on individual people - a classical example of pairing - where each person will have a different baseline.)

5. Since all genetic manipulations rely on RNAi, it would be helpful to confirm some of the observed genotypes with an independent RNAi line. E. g., is the AMPK gamma subunit required that senses nucleotides?

We agree with the reviewer. Although many of the conclusions come from complementary approaches, a few of them rely on a single RNAi. So we went through the manuscript and identified the conclusions that hinge on a single RNAi, and then added an additional RNAi:

- For experiments using Adgf-A RNAi, we also have a complementary UAS-Adgf-A overexpression experiment which shows the opposite phenotype, as well as adenosine feeding experiments.

- For the Adk's, single knockdowns of Adk2 or Adk3 with separate RNAis show the same phenotype.

- For AMPK, as suggested by the reviewer, we have now tested the knockdown of the AMPK gamma subunit in wing discs of *spokⁱ* larvae fed 20E to see if this enables discs to proliferate at 96h (Reviewer Fig. 14. ED Fig. 5b). This is indeed the case. Note that the level of proliferation upon knockdown of AMPK γ is not as high as for AMPK α , likely due to the fact that the knockdown of AMPK α is more efficient than the knockdown of AMPK γ , quantified by Q-RT-PCR in wing discs (ED Fig. 5c).

Reviewer Figure 16: Knockdown of AMPK gamma in wing discs of animals lacking endogenous ecdysone biosynthesis (*spokⁱ*) enables wings to proliferate after 96h of 20E feeding, when they would normally stop. (Top) representative images. (Bottom) quantification EdU.

- Most experiments with EcR knockdown to induce target gene expression are complemented by equivalent results obtained with ecdysone feeding. The main result that we cannot show with ecdysone feeding is that ecdysone signaling in intestinal enterocytes regulates Adgf-A expression in enterocytes. We therefore now tested a second EcR RNAi line (Reviewer Figure 17, ED Fig. 3d), which confirms that EcR knockdown in enterocytes inhibits Adgf-A expression.

Reviewer #3

Zhang et al explore the interplay between ecdysone signalling and adenosine metabolism in regulating imaginal tissue growth in *Drosophila*. The authors provide compelling evidence that the steroid hormone ecdysone affects imaginal tissue proliferation both autonomously and non-autonomously through its effects on intestinal enterocytes. In particular, the authors show that ecdysone inhibits the production by intestinal enterocytes of Adgf-A a secreted enzyme that in turn depletes circulating adenosine pools. Adenosine depletion allows imaginal disc cells to proliferate by relieving the activation of the metabolic sensor AMPK, which is allosterically activated by adenosine conversion to AMP inside wing disc cells. Excitingly, this "remote-control" of imaginal disc cell proliferation by ecdysone via an intestinal relay is sensitive to dietary amino acids, providing a link between nutrient uptake and hormonal control of proliferation. The fact that ecdysone promotes imaginal disc cell proliferation tissue autonomously (shown here and in previous publications) and inhibits it with a delay via enterocytes and adenosine (this manuscript) opens the exciting possibility suggested by the authors that ecdysone pulses license successive pulses of growth throughout the developing animal that could be responsible for the coupling between growth and developmental progression. This studies therefore advances our understanding of developmental growth and opens up a rich area for future studies.

The data are generally compelling and well presented. There are a few mainly technical issues to be addressed in order to bolster the authors' conclusions as detailed below. With these points solidly explored, I believe this manuscript would have

substantial impact in the field and potentially wide implications for our understanding of metabolic scaling, as well as tumour biology or other developmental growth disorders.

We thank the reviewer for the positive evaluation, and for the constructive suggestions below, which we have tried to fully address.

1) Specificity of GAL4 Drivers

The authors' tissue-specific knockdown experiments explore the non-autonomous effects of ecdysone signalling. However, as is usually the case some of the GAL4 driver used are not completely specific in their expression patterns. It is important to strengthen the data implicating enterocytes as the source of Adgf-A, and exclude the ring gland and hemocytes in this context, particularly since prior work suggests that ring gland/hemocyte-expressed Adgf-A is sufficient to rescue developmental defects in Adgf-A mutants (Dolezal et al., 2005).

We therefore suggest the following experiments:

- Use the alternative enterocyte driver mex1-GAL4 (BL91368) to repeat the experiments in Fig. 1f and 4a.

We have now repeated the experiments in Fig. 1f and 4a with the Mex1-GAL4 driver. These data are now shown in ED Fig. 2c and ED Fig. 4e (and below as Reviewer Figures 18 and 19), and they completely recapitulate the results originally obtained with Myo1A-GAL4.

- Add the general ring gland driver P0206-GAL4 and a hemocyte driver (hml-GAL4 or eater-GAL4) to the panel in ED Fig. 2.

We have now added P0206-GAL4 and hml-GAL4 to the experiments in ED Fig.2 (now shown as ED Fig. 2b and below in Reviewer Fig. 20). Neither GAL4 driver causes the wing disc to stop proliferating, indicating that the effect is not mediated by the ring gland or by hemocytes, and consistent with the gut being the key organ mediating this effect.

Ideally the authors could also check that MyoIA and mex1-GAL4 are not significantly active in the L3 ring gland.

We have now crossed Myo1A-GAL4 and Mex1-GAL4 to UAS-GFP and imaged the L3 ring gland. Compared to the gut, there is hardly any visible GFP in the ring gland. Nonetheless, if we boost the laser and do a long exposure, we can detect a very weak signal, which is even weaker than the background expression in the neighboring brain (ED Fig. 2d and Reviewer Figure 21). We have included these data in the manuscript, but we believe the best data excluding that the ring gland mediates this effect are the experiments where we knock down EcR in the ring gland with phm-GAL4 or P0206-GAL4, both of which show no effect (ED Fig. 2).

2) Mechanism of intestinal ecdysone signalling

Many possible mechanisms could explain how Adgf-A expression is activated by dietary amino acids in enterocytes, but the simplest to test would be mTORC1-dependent. The authors could increase/decrease mTORC1 activity in ECs (Rheb/Tsc1/2 overexpression) and test the effect on Adgf-A expression.

We thank the reviewer for this suggestion. We find that constitutively activating mTORC1 by knocking down TSC2, or constitutively inactivating mTORC1 by knocking down Rheb or Raptor in enterocytes does not blunt the induction of Adgf-A in response to amino acids (Reviewer Fig. 22 top panel, ED Fig. 6f). Instead, knockdown of either GCN2 or the downstream transcription factor ATF4 strongly blunts this induction (lower panel Reviewer Figure 11, ED Fig. 6g). We think this is interesting because it provides a physiological function for GCN2, which senses the absence of any amino acid. These results add significant new mechanistic insight to the story and we thank the reviewer for having suggested this experiment.

Reviewer #1 (Remarks to the Author):

The authors have provided a thorough and careful revision of the manuscript, addressing the concerns raised. The additional experiments substantially strengthen the conclusions and provide strong support for the proposed model. The inclusion of GCN2/ATF4-dependent regulation of Adgf-A expression further clarifies how dietary amino acids control this pathway. The statistical analyses have also been carefully described. That said, I note that several of the figures still lack statistical annotations, such as those showing growth rates and pupariation timing (ED Fig. 6), ED Fig. 7e where ecdysone-inducible gene expression is measured, and Reviewer Fig. 9 presenting Adgf-A mRNA levels under different dietary conditions. Nevertheless, these points do not affect the overall conclusions of the manuscript but should be addressed. In summary, the authors have addressed the previous concerns in a satisfactory manner. The new data convincingly support the central claims, and the revised manuscript is suitable for publication after minor attention to the remaining points regarding missing statistical annotations.

We thank the reviewer for the supportive comments. We have added statistical annotations to the following figures: ED Fig.6, ED Fig.7a and Reviewer Fig.9 (seen below as Reviewer Fig.1).

Reviewer #2 (Remarks to the Author):

in this manuscript, Zhao et al develop a system to distinguish a cell autonomous and a non-autonomous effect of ecdysone on wing disk growth, identifying a second growth-inhibitory effect that is non-autonomous. This effect is mediated by gut enterocytes that secrete Adgf to modulate hemolymph levels of adenosine, which had previously been identified as a growth inhibitor for wing disks. They suggest that adenosine is taken up intracellularly and phosphorylated to yield AMP. AMP then can activate the AMP-dependent kinase, AMPK, which in turn inhibits TORC1 to stunt growth. In addition to ecdysone, adenosine deaminase expression in the gut is regulated by dietary cues, and changing expression levels there has effects on developmental timing at the whole animal level.

In my original review, I asked to assess whether Adks can induce real AMP concentration changes in the wing disc, to provide further evidence that AMPK is upstream of TOR in the Adgf pathway, to assess potential roles for nucleoside transporters in adenosine uptake, to reassess their use of statistic tests and to confirm key experiments with independent RNAi lines. The authors have addressed these questions convincingly, and the experiments support the proposed model. I am particularly impressed that one can see the effect of Adk knockdown on AMP concentrations.

We thank the reviewer for this positive evaluation. We are pleased that our revised manuscript satisfied the reviewer.

Reviewer #3 (Remarks to the Author):

The authors have addressed my comments in the revised version and I am happy to support publication in Nature Cell Biology.

We thank the reviewer for their positive feedback and support for publication.